# EFFI-CODE: UNLEASHING CODE EFFICIENCY IN LANGUAGE MODELS

## ABSTRACT

As the use of large language models (LLMs) for code generation becomes more prevalent in software development, it is critical to enhance both the efficiency and correctness of the generated code. Existing methods and models primarily focus on the correctness of LLM-generated code, ignoring efficiency. In this work, we present EFFI-CODE, an approach to enhancing code generation in LLMs that can improve both efficiency and correctness. We introduce a Self-Optimization process based on Overhead Profiling that leverages open-source LLMs to generate a high-quality dataset of correct and efficient code samples. This dataset is then used to fine-tune various LLMs. Our method involves the iterative refinement of generated code, guided by runtime performance metrics and correctness checks. Extensive experiments demonstrate that models fine-tuned on the EFFI-CODE show significant improvements in both code correctness and efficiency across task types. For example, the pass@1 of DeepSeek-Coder-6.7B-Instruct generated code increases from **43.3%** to **76.8%**, and the average execution time for the same correct tasks decreases by **30.5%**. EFFI-CODE offers a scalable and generalizable approach to improving code generation in AI systems, with potential applications in software development, algorithm design, and computational problem-solving.

## 1    INTRODUCTION

Large language models (LLMs) have recently made significant strides across various tasks (OpenAI, 2023; Anil et al., 2023; Anthropic, 2024; Meta, 2024), including code-related applications like code completion (Chen et al., 2021; Austin et al., 2021), debugging (Haque et al., 2022; Chen et al., 2023), and translation(Rozière et al., 2020; Ahmad et al., 2023). These advanced tools have been seamlessly integrated into popular development environments, enhancing developer productivity by providing intelligent code recommendations based on natural language instructions.

Before deploying LLMs into integrated development environments (IDEs) as tools, it is crucial to ensure that the generated code meets the required efficacy standards. To address this, researchers have explored various datasets to fine-tune LLMs, thereby improving the efficacy of LLM-generated code (Ouyang et al., 2022; Wei et al., 2022). For example, Code Alpaca (Chaudhary, 2023) utilized the Self-Instruct framework (Wang et al., 2023) to synthesize data, while WizardCoder (Luo et al., 2024) employed the Evol-Instruct technique (Xu et al., 2024) to generate heuristic prompts for diverse solutions. Additionally, OSS-Instruct (Wei et al., 2024) created new coding problems using open-source snippets with LLMs, and Octopack (Muennighoff et al., 2024) focused on curating high-quality Git commit messages that resemble natural language instructions. These fine-tuning efforts have led to increased correctness in LLM-generated code.

However, our observation is that existing works primarily focus on enhancing the correctness of LLM-generated code while neglecting to optimize its efficiency. As a result, the efficiency of such code often falls short compared to canonical solutions written by human developers. Recent studies (Shi et al., 2024; Niu et al., 2024; Du et al., 2024; Huang et al., 2024a) also point out that LLM-generated code typically exhibits lower efficiency in terms of execution time and memory usage. For instance, on the EffiBench benchmark (Huang et al., 2024b), even the most advanced LLMs, such as GPT-4-Turbo, produced less efficient code, with average and worst-case execution times being 1.69 and 45.49 times longer than those of canonical solutions, respectively.

Efficiency is crucial because inefficient code consumes more computational resources, leading to higher energy consumption and increased operational costs. This is particularly important in the context of sustainability, as the demand for computing power continues to grow, and reducing the environmental impact of large-scale computations becomes a pressing concern. Furthermore, inefficient code may be impractical for use in resource-constrained environments, such as mobile devices or embedded systems, where both energy and processing power are limited. This underscores the urgent need to develop new methods that can enhance both the **correctness** and **efficiency** of LLM-generated code.

In this paper, we introduce the dataset EFFI-CODE, aimed at fine-tuning LLMs to improve both code efficiency and correctness. We begin by aggregating source code from eight existing open-source datasets available on the Hugging Face platform. This is followed by a rigorous preprocessing and cleaning process, coupled with the generation of test cases for each task to evaluate code efficiency. The cleaned code is executed using test cases to profile memory usage and execution time Huang et al. (2024a). Through a self-optimization process based on these profiles Huang et al. (2024a), we iteratively refine the code over five optimization cycles. The resulting optimized code, along with its associated metadata, forms the foundation of our fine-tuning dataset, EFFI-CODE, which serves as a high-quality resource for training LLMs to generate more efficient code while ensuring correctness.

Extensive experiments on HumanEval (Chen et al., 2021) and EffiBench (Huang et al., 2024b) demonstrate that fine-tuning LLMs with EFFI-CODE improves both correctness and efficiency. For example, the fine-tuned DeepSeek-Coder-6.7B (DeepSeekAI, 2023) increases the pass@1 from 43.3% to 76.8% on HumanEval, while also reducing the average execution time from 0.59 seconds to 0.41 seconds — representing a 30.5% reduction in execution time overhead. Compared to PIE (Shypula et al., 2024), which increases the pass@1 from 12.2% to 19.5% on HumanEval, the pass@1 of CodeLlama-7B (Rozière et al., 2023) fine-tuned with EFFI-CODE further increases to 37.8%. In addition, EFFI-CODE decreases the execution time by 7.1% while PIE decreases it by 4.8%. We will fully open-source EFFI-CODE, the source code, and model weights to facilitate research. To conclude, this paper makes the following contributions:

- We provide a framework to inspire researchers to construct code generation datasets containing efficient solutions for each code generation task, which is versatile and can be adapted to different programming languages and leverage various existing data sources. Unlike some other code generation datasets that rely on powerful models (e.g., GPT-4), our framework can be implemented only using open-sourced LLMs. The framework provides a systematic method for researchers to enhance existing datasets or create new ones focused on code efficiency across different languages and domains.

- Based on our proposed framework, we release the Effi-Code dataset. To the best of our knowledge, it is the first instruct tuning dataset that focuses on improving the efficiency of LLM-generated code. The primary purpose of Effi-Code is to instruct and fine-tune LLMs to ensure that the LLM-generated code is more efficient.

- We use Effi-Code to fine-tune widely used LLMs and will release these models on the Hugging Face website in the final version. Different from existing datasets that are used to finetune the LLMs to improve the pass@1 of LLM-generated code, our evaluation results demonstrate that both the pass@1 and the efficiency results would be improved for LLMs finetuned on our Effi-Code dataset.

## 2 RELATED WORKS

Instruction tuning has proven effective in enhancing the usability and overall performance of LLMs across various language tasks (Ouyang et al., 2022; Wei et al., 2022; Zhao et al., 2024). This approach has been extended to the domain of code generation. The core challenge is the acquisition of high-quality instructional data, which is often labor-intensive. To address this, recent research has focused on developing methods to generate synthetic instruction data. Studies have shown that textbook-quality synthetic data alone can improve a model's coding and reasoning capabilities (Gunasekar et al., 2023; Li et al., 2023b). One early effort was Self-Instruct (Wang et al., 2023), which utilized LLMs to generate synthetic instruction-response pairs using carefully crafted prompts. The same LLM was then instruction-tuned on this synthetic data. Code Alpaca (Chaudhary, 2023) applied the

Self-Instruct approach with GPT models, tailoring it specifically for code generation, editing, and optimization tasks. Building upon this, WizardCoder (Luo et al., 2024) adapted the Evol-Instruct technique (Xu et al., 2024) to the coding domain by designing heuristic prompts to create more complex and diverse synthetic data. OSS-Instruct (Wei et al., 2024) took a different approach by leveraging LLMs to automatically generate new coding problems inspired by random code snippets from open-source repositories. In contrast, Octopack (Muennighoff et al., 2024) focused on collecting and filtering high-quality Git commit messages that resemble natural language instructions. While these existing methods primarily emphasize generating correct code, EFFI-CODE explores the use of fine-tuning to improve code efficiency. Our method is orthogonal to existing synthetic techniques, offering the potential for combination to further enhance the coding capabilities of LLMs.

## 3 EFFI-CODE: FINE-TUNING FOR EFFICIENCY

In this section, we provide a det ailed pipeline for constructing the dataset EFFI-CODE for fine-tuning. Specifically, we first collect source code from eight existing open-source datasets available on the HuggingFace platform[1]. To ensure the quality and usability of the collected data, we use several filtering strategies, such as filtering tasks that are not algorithmic tasks and do not require efficiency optimization[2]. In addition, we also generate test cases for each task to ensure that we can measure the efficiency of each task's source code.

Next, we execute the cleaned source code using the generated test cases to profile the memory usage and execution time for each task. Then, we use Self-Optimization based on these overheAd Profiles (SOAP; Huang et al. 2024a), which iteratively refines the code over five optimization cycles to generate an efficient solution for each task in the collected tasks. Finally, we process the optimized code and the associated metadata to create our final fine-tuning dataset, EFFI-CODE, which is carefully curated to provide a high-quality resource for training models to generate more efficient code while maintaining correctness.

### 3.1 SOURCE DATA COLLECTION

To construct a high-quality dataset to improve code efficiency, the first important step is to collect a large number of code task candidates, which will be used for further processing. In our experiments, we collected code candidates from several existing code generation tasks. As shown in Table 1, the collected datasets include CodeFeedback-Filtered-Instruction (CodeFeed; MAP 2023), Tested-143k-Python-Alpaca (Alpaca; Vezora 2023), Glaive-Code-Assistant (Glaive; Computer 2023), Magicoder-Evol-Instruct-110K (Evol-Ins; UIUC 2023a), Dolphin-Coder (Dolphin; Computations 2023), Magicoder-OSS-Instruct-75K (Oss-Ins; UIUC 2023b), Self-OSS-Instruct-SC2-Exec-Filter-50K (Self-Oss; BigCode 2023), and Apps (Hendrycks et al., 2021).

### 3.2 PRE-SOAP CONSTRUCTION

Before the SOAP stage (Section 3.3), we construct the correct solutions and unit test cases for the collected data. To create a well-structured dataset for the SOAP process, we follow the steps below to filter and process tasks from our collected candidates:

**Convert code into functions (Step 1):** The first step in our experiments is to convert the Python source code for tasks that are not initially in function format into a function representation and filter out tasks that are not written in Python. For example, in the original solutions provided by the APPS dataset, some task solutions are not at the function level. In this setup, we convert these solutions into function-level representations. Additionally, since the test cases for these tasks are not in the unit test case format, we also convert them into unit test cases using the following format: `assert function_name(inputs) == outputs`.

**Filter tasks with risky operations (Step 2):** In our experiments, some datasets are generated based on Language Models (LLMs), where they first require an LLM (e.g., GPT-3.5-turbo) to generate task descriptions and then generate source code based on those descriptions. As the source code

---

[1] `https://huggingface.co/docs/datasets/index`

[2] Data decontamination was not included in the filtering process as most of the tasks we collected have been decontaminated, such as OSS-Instruct (UIUC, 2023b).

Table 1: The statistics of the dataset construction process. We start with a large pool of tasks from various datasets and apply a series of filtering steps to create a high-quality dataset for fine-tuning. In the pre-SOAP cleaning phase, we convert the code into functions (Step 1), filter tasks with risky operations (Step 2), construct test cases (Step 3), and filter non-algorithmic tasks (Step 4). After applying SOAP to optimize the code, we perform post-SOAP cleaning by filtering tasks not addressed by the teacher model (Step 5) and tasks without efficient solutions (Step 6). The resulting dataset contains tasks with optimized solutions that demonstrate significant efficiency improvements.

| Dataset | CodeFeed | Alpaca | Glaive | Evol-Ins | Dolphin | Oss-Ins | Self-Oss | Apps |
|---|---|---|---|---|---|---|---|---|
| Initial Size | 156526 | 143327 | 136109 | 111183 | 109118 | 75197 | 50661 | 5000 |
| **Pre-SOAP** | | | | | | | | |
| Step 1 | 76534 | 121810 | 46422 | 40285 | 21154 | 40459 | 50660 | 2731 |
| Step 2 | 15180 | 33262 | 16700 | 10078 | 4938 | 4961 | 15477 | - |
| Step 3 | 13953 | 29746 | 14703 | 9061 | 4318 | 4353 | 3183 | - |
| Step 4 | 3704 | 12320 | 94 | 3136 | 3892 | 388 | 2328 | 2183 |
| **Post-SOAP** | | | | | | | | |
| Step 5 | 3691 | 12293 | 94 | 3133 | 3870 | 388 | 2316 | - |
| Step 6 | 1387 | 2920 | 32 | 1250 | 1958 | 76 | 827 | 1001 |

generated by LLMs is not evaluated locally, some tasks with risky operations (e.g., deleting system files) may not be filtered out. To address this, we feed all tasks into GPT-3.5-turbo and require it to analyze whether the source code contains any risky operations. We then remove tasks that are labeled as containing risky operations.

**Construct test cases (Step 3):** In our experiments, most tasks do not have existing test cases[3]. To address this, we use GPT-3.5-turbo to construct test cases by feeding the task description and source code into the model and requiring it to generate test cases for our experiments. After that, we analyze whether each test case generated by GPT-3.5-turbo is correct and then filter out incorrect test cases and tasks that do not have correct test cases. To determine the correctness of the test cases generated by GPT-3.5-turbo, we execute each test case individually with the initial solution provided for each task in our collected candidate tasks. These initial solutions are usually correct but do not have efficiency optimization. We check whether any errors are raised during the execution of each test case with the initial solution. In other words, we verify if the test case passes the initial solution. Since the initial solutions are correct, we treat the test cases that pass the canonical solution as correct. On the other hand, test cases that do not pass the canonical solution are filtered out. By using the canonical solution as a reference, we can effectively assess the correctness of the generated test cases and ensure that only valid test cases are retained for further analysis.

**Filter non-algorithmic tasks (Step 4):** Finally, we filter out tasks that do not involve algorithms. We define a task as 'non-algorithmic' if it does not require a specific algorithm or computational steps to solve. non-algorithmic tasks might involve coding but do not require complex algorithmic reasoning. Instead, they might focus on straightforward implementation or basic syntax usage. For example, an algorithmic task may be *Implement a function to find the longest palindromic substring in a given string.* This requires an understanding of dynamic programming and string manipulation algorithms. While a non-algorithmic task may be *Write a function to print 'Hello, World!'.* This is a clear example of routine implementation without algorithmic challenges. The primary motivation for filtering out non-algorithmic tasks is to ensure that our dataset focuses on problems that assess algorithmic thinking and coding skills. By excluding tasks that do not require algorithmic problem-solving, we maintain the coherence and relevance of our dataset to the intended purpose of evaluating AI models' coding abilities. To identify and filter out non-algorithmic tasks, we provide the task description and the canonical solution to GPT-3.5-turbo and request it to analyze whether the given task is an algorithmic task based on our provided definition. GPT-3.5-turbo is instructed to return a binary classification (True or False) based on its analysis. Tasks classified as False are considered non-algorithmic and are subsequently removed from our candidate tasks.

---

[3]Some datasets do not generate test cases as they do not need to check the correctness of the source code.

## 3.3 SELF-OPTIMIZATION BASED ON OVERHEAD PROFILE (SOAP)

To optimize the source code in our collected tasks, we employ the **S**elf-**O**ptimization based on overhe**A**d **P**rofile (SOAP; Huang et al. 2024a) to optimize the efficiency of the source code. For each task in our dataset, we execute the source code using the generated test cases and profile the execution time and memory usage for each line of code using the `line_profiler` and `memory_profiler` libraries in Python. The profiling results, along with the original source code and task description, are then fed into DeepSeek-Coder-V2-Lite (Zhu et al., 2024), which analyzes the profiles to identify performance bottlenecks and inefficiencies in the code. The model applies various optimization techniques to refine the code for better efficiency, and the optimized code is validated against the provided test cases to ensure its functional correctness. This process is repeated for a predefined number of optimization iterations. By applying SOAP to our collected tasks, we create a dataset of optimized source code that demonstrates improved efficiency compared to the original code. This dataset serves as a valuable resource for training models to generate more efficient code and for understanding the effectiveness of LLM-driven code optimization techniques.

## 3.4 POST-SOAP CLEANING

After generating efficient source code based on SOAP, we then filter tasks in our candidate pool to enable our fine-tuning process.

**Filtering tasks not addressed by the Teacher Model (Step 5):** As mentioned in Section 3.3, we use DeepSeek-Coder-V2-Lite to construct more efficient solutions for our candidate tasks. However, some tasks are not addressed by DeepSeek-Coder-V2-Lite, which means that we cannot obtain "efficient" solutions for these tasks in our experiments. To maintain the quality and consistency of our dataset, we remove these unaddressed tasks from our candidate pool. This filtering step ensures that all tasks in our dataset have been successfully optimized by the teacher model, providing a reliable foundation for the fine-tuning process.

**Filtering tasks without efficient solutions (Step 6):** We define a solution as inefficient if it exhibits suboptimal execution time or memory usage compared to the initial solution (solution provided by the collected dataset) for the given task. The criteria for determining inefficiency are based on the potential for improvement in terms of execution time and memory usage after applying optimization techniques. Consider a task where the goal is to sort an array. An inefficient solution uses Bubble Sort, which has a time complexity of $O(n^2)$, as opposed to an efficient solution like Quick Sort with an average time complexity of $O(n \log n)$.

Despite the application of SOAP (Huang et al., 2024a), some tasks may not yield more efficient solutions due to limited optimization potential, even though they are algorithmic tasks. In such cases, the SOAP process may not be able to generate solutions that significantly improve upon the original code in terms of efficiency. To ensure that our dataset focuses on tasks with meaningful optimization potential, we filter out these tasks from our experiments. To identify and filter out tasks with inefficient solutions, we employ a two-step process. First, we use self-optimization to require DeepSeek-Coder-V2-Lite to improve the efficiency of the code solutions, which aims to improve the efficiency of the code by making optimizations such as reducing redundant computations or improving data structures. We run DeepSeek-Coder-V2-Lite for five iterations and analyze whether the efficiency of the code has improved based on metrics such as execution time and memory usage. If the efficiency does not show improvement after these iterations, we consider the task to have an inefficient solution and remove it from our candidate tasks. We acknowledge that there may be cases where the initial code is already efficient, and the lack of improvement after optimization does not necessarily indicate an inefficient solution. However, detecting such cases would require significant manual effort to analyze each task individually. To maintain a consistent and automated approach, we opted to remove all tasks that did not show efficiency improvement after the optimization process, which proved to still perform very well in our evaluation.

The post-SOAP cleaning process plays a crucial role in refining our candidate tasks and creating a high-quality dataset for fine-tuning. By filtering out tasks that are not addressed by the teacher model and those without significant efficiency improvements, we ensure that our final dataset consists of tasks with optimized solutions that demonstrate a notable enhancement in performance. This curated dataset serves as a valuable resource for training models to generate efficient code and for advancing the field of code optimization using LLMs.

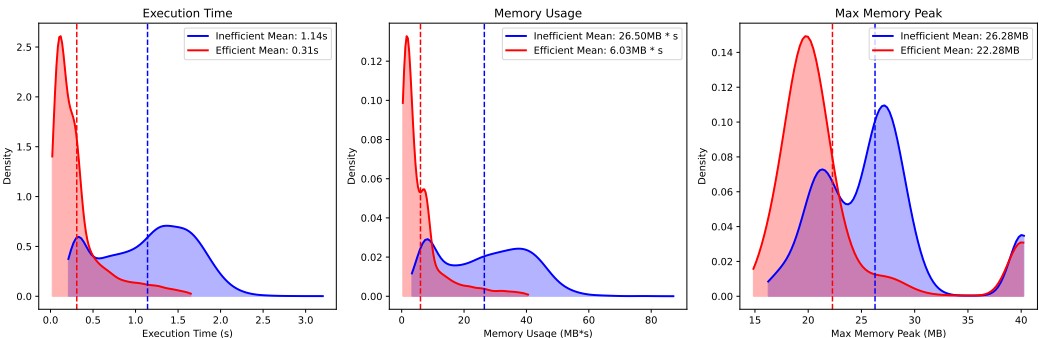

Figure 1: Efficiency distribution of the dataset. The figure shows the distribution of execution time, memory usage, and max memory peak for both inefficient (task-provided solution) and efficient solutions in the EFFI-CODE. The inefficient solutions have higher overheads for all three metrics compared to the efficient solutions.

## 3.5 EVALUAITON METRICS

Following Huang et al. (2024b), we evaluate the effectiveness of EFFI-CODE fine-tuned LLMs using two key aspects: correctness and efficiency of the LLM-generated code. Our metrics are outlined as:

- **Execution Time (ET)**: Measures the time taken for code execution.
- **Max Memory Usage (MU)**: Assesses the peak memory requirement during code execution.
- **Total Memory Usage (TMU)**: Evaluates the overall memory usage throughout code execution.
- **Normalized Metrics**: The metrics contains NET (Normalized Execution Time), NMU (Normalized Max Memory Usage), and NTMU (Normalized Total Memory Usage). They are our primary metrics for assessing efficiency, measuring how efficient/inefficient the LLM-generated code is compared with the human-written canonical solution for ET, MU, and TMU.
- **Correctness**: We assess the correctness of LLM-generated code using the pass@1 metric with greedy decoding, following the approach of existing works.

## 3.6 DATASET STATISTICS

As shown in Table 1, coding problems in EFFI-CODE have been collected from eight datasets, resulting in 9,451 tasks. The initial pool of tasks was quite large, with over 780,000 tasks across the eight datasets. However, through our rigorous cleaning processes, we carefully filtered and refined the tasks to create a high-quality dataset for fine-tuning. The final EFFI-CODE contains 9,451 tasks, with contributions from each of the eight datasets as follows: 1,387 tasks from CodeFeedback, 2,920 tasks from Alpaca, 32 tasks from Glaive, 1,250 tasks from Evol-Ins, 1,958 tasks from Dolphin, 76 tasks from Oss-Ins, 827 tasks from Self-Oss, and 1,001 tasks from Apps.

Figure 1 illustrates the efficiency distribution of the dataset for three key metrics: execution time, memory usage, and max memory peak, which compares the distribution of these metrics for both inefficient (canonical solutions provided by the eight datasets) and efficient solutions in the EFFI-CODE. For execution time, the inefficient solutions have a mean value of 1.14s, while the efficient solutions have a significantly lower mean of 0.31s, which indicates that the optimization process has successfully reduced the execution time of the code, resulting in more efficient solutions. Similarly, the memory usage and max memory peak also show a notable difference between inefficient and efficient solutions. For example, inefficient solutions have a mean memory usage of 26.50MBs, while the efficient solutions have a much lower mean of 6.03MBs.

The efficiency distribution visualization highlights the effectiveness of the optimization process in creating more efficient solutions across all three metrics. By carefully curating tasks through the multi-step cleaning process and applying SOAP optimization, we have created a dataset that serves as a valuable resource for training models to generate efficient code. EFFI-CODE provides a diverse range of optimized coding problems, enabling researchers and practitioners to advance the field of code optimization using LLMs.

Table 2: Code efficiency and pass@1 of LLMs trained with EFFI-CODE. The percentage in the brackets indicates the extent of the reduction for each respective item. `Overlap` means the percentage of correct tasks addressed by both EFFI-CODE finetuned LLM and original LLM in total tasks of the dataset. We provide a case example in Figure 3 to demonstrate how EFFI-CODE fine-tuned LLM improves the efficiency of LLM-generated code.

| Model | ET (s) ↓ | NET ↓ | MU (Mb) ↓ | NMU ↓ | TMU (Mb*s) ↓ | NTMU ↓ | Overlap (%) ↑ | Pass@1 (%) ↑ |
|---|---|---|---|---|---|---|---|---|
| | | | | HumanEval | | | | |
| DeepSeek-Coder-6.7b-base | 0.89 | 2.07 | 67.50 | 1.00 | 56.66 | 1.96 | 7.3 | 7.3 |
| + SFT (Ours) | 0.71 (20.2%) | 1.14 (44.9%) | 67.50 (0.0%) | 1.00 (0.0%) | 53.09 (6.3%) | 1.16 (40.8%) | 7.3 | 59.8 |
| DeepSeek-Coder-6.7b-instruct | 0.59 | 2.07 | 63.48 | 0.99 | 24.42 | 2.08 | 39.0 | 43.3 |
| + SFT (Ours) | 0.41 (30.5%) | 1.19 (42.5%) | 63.48 (0.0%) | 0.99 (0.0%) | 19.96 (18.3%) | 1.36 (34.6%) | 39.0 | 76.8 |
| Qwen2.5-Coder-7B | 0.59 | 1.95 | 61.95 | 0.99 | 24.29 | 1.83 | 56.1 | 63.4 |
| + SFT (Ours) | 0.40 (32.2%) | 1.01 (48.2%) | 61.96 (-0.0%) | 0.99 (0.0%) | 18.74 (22.8%) | 1.02 (44.3%) | 56.1 | 79.9 |
| Qwen2.5-Coder-7B-Instruct | 0.74 | 2.72 | 62.81 | 1.00 | 35.43 | 3.15 | 51.2 | 54.3 |
| + SFT (Ours) | 0.51 (31.1%) | 1.68 (38.2%) | 62.77 (0.1%) | 1.00 (0.0%) | 28.01 (20.9%) | 2.24 (28.9%) | 51.2 | 84.8 |
| | | | | EffiBench | | | | |
| DeepSeek-Coder-6.7b-base | 0.44 | 2.61 | 57.24 | 1.26 | 54.57 | 7.94 | 7.3 | 8.5 |
| + SFT (Ours) | 0.29 (34.1%) | 2.08 (20.3%) | 50.58 (11.6%) | 1.00 (20.6%) | 17.25 (68.4%) | 2.79 (64.9%) | 7.3 | 57.6 |
| DeepSeek-Coder-6.7b-instruct | 0.14 | 1.00 | 38.36 | 1.00 | 4.21 | 0.97 | 1.0 | 1.3 |
| + SFT (Ours) | 0.13 (7.1%) | 0.93 (7.0%) | 38.31 (0.1%) | 1.00 (0.0%) | 4.01 (4.8%) | 0.92 (5.2%) | 1.0 | 51.6 |
| Qwen2.5-Coder-7B | 0.26 | 1.79 | 38.06 | 1.01 | 18.30 | 2.74 | 44.2 | 50.1 |
| + SFT (Ours) | 0.21 (19.2%) | 1.45 (19.0%) | 38.15 (-0.2%) | 1.01 (0.0%) | 15.88 (13.2%) | 1.70 (38.0%) | 44.2 | 63.9 |
| Qwen2.5-Coder-7B-Instruct | 0.44 | 3.96 | 28.62 | 1.00 | 10.17 | 5.43 | 3.2 | 3.3 |
| + SFT (Ours) | 0.43 (2.3%) | 3.88 (2.0%) | 28.59 (0.1%) | 1.00 (0.0%) | 10.10 (0.7%) | 5.37 (1.1%) | 3.2 | 61.0 |

## 4 EXPERIMENT

**Datasets and Models** In our experiments, we evaluate the efficiency and correctness of LLM-generated code on two code generation benchmarks, i.e., HumanEval and EffiBench. We fine-tune four open-source LLMs with EFFI-CODE, including DeepSeek-Coder-6.7B base and instruct model (DeepSeekAI, 2023), Qwen2.5-Code-7B base and instruct model (Hui et al., 2024).

**Fine-tuning Setup** We use Llama-factory (Zheng et al., 2024) to fully fine-tune all LLMs with the same setup and train the models using EFFI-CODE. The maximum sequence length is set to 2048 tokens. We use a batch size of 128 and set the learning rate to 5e-6 with a cosine learning rate scheduler and a warmup ratio of 0.03. We fine-tune all LLMs for 4 epochs under the bf16 data type.

**Prompt Template** For all experiments, we use the inference prompt provided by DeepSeek-Coder for both fine-tuning and inference. The detailed template can be found in Appendix A.3.

### 4.1 MAIN RESULTS

The evaluation results of EFFI-CODE are shown in Table 2, where we can observe that EFFI-CODE can improve both the efficiency and the correctness (pass@1) for LLM-generated code in most of the experiments across HumanEval and EffiBench.

**HumanEval** We observe that all LLMs achieve better efficiency and higher correctness after being fine-tuned with EFFI-CODE. For instance, the pass@1 of DeepSeek-Coder-6.7B-Instruct on HumanEval is 43.3%. However, the fine-tuned DeepSeek-Coder-6.7B-Instruct achieves a pass@1 of 76.8% for the same dataset. Furthermore, the average execution time (ET) for all correct tasks addressed by both the initial and fine-tuned model generated by DeepSeek-Coder-6.7B-Instruct is 0.59 (s), while it decreases to 0.41 (s) for EFFI-CODE fine-tuned DeepSeek-Coder-6.7B-Instruct, resulting in a 30.5% reduction in average execution time.

**EffiBench** As shown in Table 2 *EffiBench*, similar to the results of the HumanEval dataset, EFFI-CODE fine-tuned LLMs increase the overall pass@1 and efficiency of the generated code. For example, the pass@1 of DeepSeek-Coder-6.7B-base achieves only 8.5%, but it reaches 57.6% when fine-tuned with EFFI-CODE. Additionally, the overhead of the LLM-generated code is significantly reduced. DeepSeek-Coder-6.7B-base requires an average of 0.44 (s) to execute its generated code. However, for the same tasks, the EFFI-CODE fine-tuned DeepSeek-Coder-6.7B-base only requires 0.29 (s), which results in an average of 34.1% decrease in execution time.

### 4.2 ABLATION STUDY

**How does the size of the fine-tuning dataset affect the effectiveness of LLM-generated code?** To investigate the impact of the fine-tuning dataset size on the effectiveness of LLM-generated code,

Table 3: Efficiency and pass@1 results for DeepSeek-Coder-6.7B-base/instruct fine-tuned on 25%, 50%, 75%, and 100% proportions of the EFFI-CODE.

| Model | ET (s) ↓ | NET ↓ | MU (Mb) ↓ | NMU ↓ | TMU (Mb*s) ↓ | NTMU ↓ | Overlap (%) ↑ | Pass@1 (%) ↑ |
|---|---|---|---|---|---|---|---|---|
| Base | 0.99 | 2.11 | 69.10 | 1.00 | 65.56 | 1.99 | 6.1 | 7.3 |
| 25 | 0.97 (2.0%) | 1.96 (7.1%) | 69.02 (0.1%) | 1.00 (0.0%) | 66.00 (-0.7%) | 1.97 (1.0%) | 6.1 | 55.5 |
| 50 | 0.98 (1.0%) | 2.03 (3.8%) | 68.78 (0.5%) | 1.00 (0.0%) | 65.03 (0.8%) | 1.90 (4.5%) | 6.1 | 54.3 |
| 75 | 0.95 (4.0%) | 1.93 (8.5%) | 68.85 (0.4%) | 1.00 (0.0%) | 64.17 (2.1%) | 1.89 (5.0%) | 6.1 | 54.3 |
| 100 | 0.80 (19.2%) | 1.13 (46.4%) | 69.01 (0.1%) | 1.00 (0.0%) | 62.14 (5.2%) | 1.15 (42.2%) | 6.1 | 59.8 |
| Instruct | 0.42 | 1.99 | 62.52 | 1.00 | 14.78 | 1.89 | 32.9 | 43.3 |
| 25 | 0.43 (-2.4%) | 2.02 (-1.5%) | 62.45 (0.1%) | 1.00 (0.0%) | 15.06 (-1.9%) | 1.91 (-1.1%) | 32.9 | 71.3 |
| 50 | 0.41 (2.4%) | 1.94 (2.5%) | 62.44 (0.1%) | 1.00 (0.0%) | 14.41 (2.5%) | 1.84 (2.6%) | 32.9 | 72.0 |
| 75 | 0.42 (0.0%) | 1.96 (1.5%) | 62.45 (0.1%) | 1.00 (0.0%) | 14.61 (1.2%) | 1.85 (2.1%) | 32.9 | 73.8 |
| 100 | 0.24 (42.9%) | 1.09 (45.2%) | 62.56 (-0.1%) | 1.00 (0.0%) | 10.10 (31.7%) | 1.15 (39.2%) | 32.9 | 76.8 |

Table 4: Efficiency and pass@1 results for different sizes of DeepSeek-Coder models.

| Model | ET (s) ↓ | NET ↓ | MU (Mb) ↓ | NMU ↓ | TMU (Mb*s) ↓ | NTMU ↓ | Overlap (%) ↑ | Pass@1 (%) ↑ |
|---|---|---|---|---|---|---|---|---|
| DeepSeek-Coder-1.3b-base | 0.51 | 1.06 | 65.61 | 1.00 | 35.67 | 1.05 | 11.0 | 12.2 |
| + SFT (Ours) | 0.50 (2.0%) | 1.05 (0.9%) | 65.37 (0.4%) | 1.00 (0.0%) | 34.65 (2.9%) | 1.03 (1.9%) | 11.0 | 43.9 |
| DeepSeek-Coder-1.3b-instruct | 0.38 | 1.14 | 63.32 | 1.00 | 21.30 | 1.21 | 34.8 | 45.7 |
| + SFT (Ours) | 0.35 (7.9%) | 1.09 (4.4%) | 63.31 (0.0%) | 1.00 (0.0%) | 19.57 (8.1%) | 1.18 (2.5%) | 34.8 | 59.1 |
| DeepSeek-Coder-6.7b-base | 0.89 | 2.07 | 67.50 | 1.00 | 56.66 | 1.96 | 7.3 | 7.3 |
| + SFT (Ours) | 0.71 (20.2%) | 1.14 (44.9%) | 67.50 (0.0%) | 1.00 (0.0%) | 53.09 (6.3%) | 1.16 (40.8%) | 7.3 | 59.8 |
| DeepSeek-Coder-6.7b-instruct | 0.59 | 2.07 | 63.48 | 0.99 | 24.42 | 2.08 | 39.0 | 43.3 |
| + SFT (Ours) | 0.41 (30.5%) | 1.19 (42.5%) | 63.48 (0.0%) | 0.99 (0.0%) | 19.96 (18.3%) | 1.36 (34.6%) | 39.0 | 76.8 |
| DeepSeek-Coder-33b-base | 1.04 | 4.44 | 57.64 | 0.93 | 56.63 | 6.75 | 16.5 | 18.9 |
| + SFT (Ours) | 0.27 (74.0%) | 1.33 (70.0%) | 61.02 (-5.9%) | 0.99 (-6.5%) | 10.81 (80.9%) | 1.61 (76.1%) | 16.5 | 66.5 |
| DeepSeek-Coder-33b-instruct | 0.49 | 1.38 | 62.51 | 0.99 | 28.18 | 1.65 | 64.0 | 70.1 |
| + SFT (Ours) | 0.39 (20.4%) | 1.11 (19.6%) | 62.56 (-0.1%) | 0.99 (0.0%) | 20.40 (27.6%) | 1.20 (27.3%) | 64.0 | 75.6 |

we conducted experiments using 25%, 50%, 75%, and 100% of the EFFI-CODE for fine-tuning the DeepSeek-Coder-6.7B-base and DeepSeek-Coder-6.7B-instruct models utilizing SFT fine-tuning. The evaluation results are shown in Table 3, providing efficiency metrics for different dataset ratios assessed from two perspectives: **individual** and **all**. The **individual** perspective evaluates the efficiency metrics for the correct code generated by both the original model and the fine-tuned model itself. **all** focuses on tasks successfully addressed by all LLMs fine-tuned with varying dataset ratios.

We can observe that as we increase the fine-tuning dataset, the pass@1 consistently improves. For example, when we increase the ratio of the fine-tuning dataset from 25% to 100%, the pass@1 of DeepSeek-Coder-6.7B-base increases from 55.5% to 59.8%, and we can also observe this trend in DeepSeek-Coder-6.7B-instruct, where the pass@1 increases from 71.3% to 76.8%. Next, we can also observe that as we increase the overall dataset ratio for fine-tuning, the efficiency metrics show a consistent trend of improvement. For instance, the average ET for DeepSeek-Coder-6.7B-base decreases from 0.99 (s) with the baseline model to 0.80 (s) with 100% of the EFFI-CODE, which results in a 19.2% decrease in execution time. Similarly, for DeepSeek-Coder-6.7B-instruct, the ET reduces from 0.42 (s) to 0.24 (s) when trained on 100% of the dataset, which highlights the effectiveness of a larger fine-tuned dataset in enhancing the efficiency of code generation.

**Is EFFI-CODE effective for different model sizes?** To evaluate the generalizability of EFFI-CODE across different model sizes during the fine-tuning process, we employed multiple versions of DeepSeek-Coder models, ranging from 1.3B to 33B parameters, for both base and instruct models. As shown in Table 4, the evaluation results demonstrate that EFFI-CODE improves performance across all model sizes. For instance, the pass@1 for the DeepSeek-Coder-1.3B-base increased significantly from 12.2% to 43.9% after fine-tuning it with EFFI-CODE, and the DeepSeek-Coder-6.7B-base also demonstrates an increase from 7.3% to 59.8%. A similar trend is observed with the instruct models, where the pass@1 for DeepSeek-Coder-1.3B-instruct improved from 45.7% to 59.1%, and for DeepSeek-Coder-6.7B-instruct, it improved from 43.3% to 76.8%. Additionally, efficiency metrics show consistent improvement across different model sizes. Specifically, the average ET for DeepSeek-Coder-33B-base decreased from 1.04 (s) to 0.27 (s) after fine-tuning, which resulted in a 74.0% decrease in execution time on average for all executed tasks. These findings suggest that as the model size increases, EFFI-CODE continues to enhance both the effectiveness and efficiency of the model-generated code.

**Whether open source model is enough to serve as a teacher model?** In our experiments, we employ DeepSeek-Coder-V2-Lite-Instruct as the teacher model to generate efficient solutions for constructing the EFFI-CODE. To assess the impact of the teacher model, we perform additional

Table 5: Comparison of code efficiency and pass@1 between different teacher models.

| Model | ET (s) ↓ | NET ↓ | MU (Mb) ↓ | NMU ↓ | TMU (Mb*s) ↓ | NTMU ↓ | Overlap (%) ↑ | Pass@1 (%) ↑ |
|---|---|---|---|---|---|---|---|---|
| DeepSeek-Coder-6.7B-base | 1.38 | 2.16 | 72.86 | 1.00 | 99.37 | 1.95 | 3.7 | 7.3 |
| Claude-3.5-Sonnet | 1.11 (19.6%) | 1.02 (52.8%) | 72.83 (0.0%) | 1.00 (0.0%) | 92.07 (7.3%) | 1.03 (47.2%) | 3.7 | 29.9 |
| GPT-4o | 1.10 (20.3%) | 0.99 (54.2%) | 72.63 (0.3%) | 1.00 (0.0%) | 91.76 (7.7%) | 0.99 (49.2%) | 3.7 | 39.0 |
| DeepSeek-Coder-V2-Lite (Ours) | 1.16 (15.9%) | 1.06 (50.9%) | 72.90 (-0.1%) | 1.00 (0.0%) | 97.47 (1.9%) | 1.08 (44.6%) | 3.7 | 59.8 |
| Instruct | 0.41 | 2.01 | 65.38 | 1.01 | 14.37 | 1.93 | 0.6 | 43.3 |
| Claude-3.5-Sonnet | 0.26 (36.6%) | 1.27 (36.8%) | 65.24 (0.2%) | 1.00 (1.0%) | 9.45 (34.2%) | 1.27 (34.2%) | 0.6 | 11.0 |
| GPT-4o | 0.20 (51.2%) | 0.98 (51.2%) | 65.13 (0.4%) | 1.00 (1.0%) | 7.34 (48.9%) | 0.98 (49.2%) | 0.6 | 9.8 |
| DeepSeek-Coder-V2-Lite (Ours) | 0.21 (48.8%) | 1.04 (48.3%) | 65.31 (0.1%) | 1.00 (1.0%) | 7.76 (46.0%) | 1.04 (46.1%) | 0.6 | 76.8 |

Table 6: Evaluation results for different teacher models of the EFFI-CODE fine-tune dataset.

| Model | ET (s) ↓ | NET ↓ | MU (Mb) ↓ | NMU ↓ | TMU (Mb*s) ↓ | NTMU ↓ | Overlap (%) ↑ | Pass@1 (%) ↑ |
|---|---|---|---|---|---|---|---|---|
| DeepSeek-Coder-6.7b-base | 0.39 | 2.00 | 62.52 | 1.01 | 12.78 | 1.85 | 1.8 | 7.3 |
| Canonical Solution | 0.42 (-7.7%) | 2.12 (-6.0%) | 62.16 (0.6%) | 1.00 (1.0%) | 14.91 (-16.7%) | 2.15 (-16.2%) | 1.8 | 15.2 |
| EFFI-CODE | 0.23 (41.0%) | 1.19 (40.5%) | 62.40 (0.2%) | 1.00 (1.0%) | 8.31 (35.0%) | 1.21 (34.6%) | 1.8 | 59.8 |
| DeepSeek-Coder-6.7b-instruct | 0.44 | 2.07 | 62.47 | 1.00 | 15.95 | 2.11 | 31.1 | 43.3 |
| Canonical Solution | 0.45 (-2.3%) | 2.11 (-1.9%) | 62.48 (-0.0%) | 1.00 (0.0%) | 16.92 (-6.1%) | 2.23 (-5.7%) | 31.1 | 57.3 |
| EFFI-CODE | 0.27 (38.6%) | 1.25 (39.6%) | 62.48 (-0.0%) | 1.00 (0.0%) | 11.81 (26.0%) | 1.45 (31.3%) | 31.1 | 76.8 |

experiments using GPT-4o-20240806 (GPT-4o) and Claude-3.5-Sonnet as alternative teacher models. The evaluation results are shown in Table 5, where we can observe that the efficient solutions generated by DeepSeek-Coder-V2-Lite-Instruct exhibit a higher pass@1 compared to those generated by GPT-4o and Claude-3.5-Sonnet. Specifically, the datasets constructed using DeepSeek-Coder-V2-Lite-Instruct fine-tuned on DeepSeek-Coder-6.7B-base achieve a 59.8% pass@1, whereas the models fine-tuned on datasets generated by the other two LLMs attain only a 39.0% pass@1. However, we can also observe that the efficiency improvement is highest for the GPT-4o-generated dataset. For example, we can observe that the ET of DeepSeek-Coder-6.7B-instruct requires 0.41 (s) to execute the correct code, while GPT-4o generated code only requires 0.20 (s) to execute for same tasks, where DeepSeek-Coder-V2-Lite-Instruct generated code also requires 0.21 (s) to execute.

**Measuring Efficiency Gains from Synthetic Code Over Original Code** In our dataset construction process, we use self-optimization with overhead profiles to generate more efficient solutions for each task and then use them for the fine-tuning process. To analyze the importance of this step, we compare the performance of LLMs fine-tuned on our self-optimized dataset with that of LLMs directly fine-tuned on the initial canonical solutions, which are usually less efficient. The evaluation results are shown in Table 6, where we can observe that directly fine-tuning LLMs with the canonical solutions provided by the dataset may not be able to improve the efficiency of LLM-generated code even though it improves the pass@1. For example, we can observe that when we directly use the dataset-provided canonical solutions to fine-tune DeepSeek-Coder-6.7B-base, the execution time increases from 0.39 (s) to 0.42 (s) for the same tasks, but it decreases to 0.23 (s) when we use EFFI-CODE's efficient solutions, which emphasizes the significance of using efficient source code for fine-tuning LLMs to generate high-performance code.

**Effectiveness with DPO fine-tuning** In Table 2, we use SFT to fine-tune LLMs with our EFFI-CODE, which raises the question of whether EFFI-CODE is also effective when using other fine-tuning techniques. To investigate this, we conduct experiments using DPO (Rafailov et al., 2024) and ORPO (Hong et al., 2024) to fine-tune DeepSeek-Coder-6.7B-instruct with EFFI-CODE. To collect preference datasets, for each task question $x$, we use our EFFI-CODE as the preferred completion $y_p$, then we use the original solution provided by each task in the datasets as dispreferred completion $y_d$, and construct the preference dataset $\mathcal{D} = \left\{ \left( x^{(i)}, y_p^{(i)}, y_d^{(i)} \right) \right\}_{i=1}^{N}$. We then fine-tune models on this dataset with two different methods.

Table 7: Code efficiency and pass@1 of DeepSeek-Coder-6.7B-instruct fine-tuned using ORPO and DPO with the EFFI-CODE.

| Model | ET (s) ↓ | NET ↓ | MU (Mb) ↓ | NMU ↓ | TMU (Mb*s) ↓ | NTMU ↓ | Overlap (%) ↑ | Pass@1 (%) ↑ |
|---|---|---|---|---|---|---|---|---|
| HumanEval | | | | | | | | |
| deepseek-coder-6.7b-instruct | 0.64 | 1.99 | 63.85 | 0.98 | 26.98 | 1.88 | 29.3 | 43.3 |
| ORPO | 0.43 (32.8%) | 0.99 (50.3%) | 63.74 (0.2%) | 0.98 (0.0%) | 20.64 (23.5%) | 1.00 (46.8%) | 29.3 | 71.3 |
| DPO | 0.44 (31.2%) | 1.00 (49.7%) | 63.78 (0.1%) | 0.98 (0.0%) | 21.11 (21.8%) | 1.02 (45.7%) | 29.3 | 55.5 |

Table 8: Code efficiency and pass@1 of DeepSeek-Coder-6.7B-instruct with EFFI-CODE with the five times execution on HumanEval.

| Model | ET (s) ↓ | NET ↓ | MU (Mb) ↓ | NMU ↓ | TMU (Mb*s) ↓ | NTMU ↓ | Pass@1 (%) ↑ |
|---|---|---|---|---|---|---|---|
| 1 | 0.47 | 1.44 | 63.17 | 0.99 | 25.10 | 1.75 | 75.6 |
| 2 | 0.46 | 1.44 | 63.12 | 0.99 | 24.98 | 1.75 | 75.6 |
| 3 | 0.47 | 1.43 | 63.17 | 0.99 | 25.17 | 1.75 | 75.6 |
| 4 | 0.47 | 1.45 | 63.15 | 0.99 | 25.01 | 1.76 | 75.6 |
| 5 | 0.46 | 1.43 | 63.15 | 0.99 | 24.84 | 1.74 | 75.6 |
| mean | 0.46 | 1.44 | 63.15 | 0.99 | 25.02 | 1.75 | 75.6 |
| std | 0.0 | 0.01 | 0.02 | 0.0 | 0.11 | 0.01 | 0.0 |

Table 9: Code efficiency and pass@1 of CodeLlama-7b-hf fine-tuned with PIE and EFFI-CODE.

| Model | ET (s) ↓ | NET ↓ | MU (Mb) ↓ | NMU ↓ | TMU (Mb*s) ↓ | NTMU ↓ | Overlap (%) ↑ | Pass@1 (%) ↑ |
|---|---|---|---|---|---|---|---|---|
| CodeLlama-7b-hf | 0.42 | 2.06 | 62.10 | 1.00 | 14.08 | 1.93 | 5.5 | 12.2 |
| PIE | 0.40 (4.8%) | 1.96 (4.9%) | 62.05 (0.1%) | 1.00 (0.0%) | 13.95 (0.9%) | 1.93 (0.0%) | 5.5 | 19.5 |
| EFFI-CODE | 0.39 (7.1%) | 1.90 (7.8%) | 61.92 (0.3%) | 1.00 (0.0%) | 13.13 (6.7%) | 1.79 (7.3%) | 5.5 | 37.8 |

The evaluation results are shown in Table 7, where we can observe that EFFI-CODE improves the performance of LLMs fine-tuned with ORPO and DPO. For example, the pass@1 of DeepSeek-Coder-6.7B-instruct increases from 43.3% to 71.3% after ORPO fine-tuning, and the average ET decreases from 0.64 (s) to 0.43 (s), which results in a 32.8% decrease in average execution time for the same tasks. Next, for DPO, we can also observe that DPO improves the performance of fine-tuned LLMs in most of the experiments. For example, the pass@1 of DeepSeek-Coder-6.7B-instruct increases from 43.3% to 55.5%, and the ET decreases from 0.64 (s) to 0.44 (s), which results in a 31.2% decrease in average execution time for the same tasks.

**Randomness** To ensure reliable model performance, we also account for variability in system conditions. Metrics like Execution Time (ET), Max Memory Usage (MU), and Total Memory Usage (TMU) might fluctuate due to factors like server workload and hardware availability, introducing noise that affects performance measurements. To demonstrate whether our results are affected by such randomness, we provide five results at different times with the mean and std for DeepSeek-Coder-6.7B-instruct in Table 8. We can observe that the results are robust as the std of the five execution times is very low for all metrics. For example, the std of ET for the five executions is 0.00.

**Comparison with PIE** To improve the efficiency of LLM-generated code, Shypula et al. (2024) propose a dataset of performance-improving edits made by human programmers consisting of over 77,000 competitive C++ programming submission pairs. To demonstrate EFFI-CODE's effectiveness, we compare the efficiency and correctness of LLM-generated code for PIE and EFFI-CODE. As PIE only releases the fine-tuned LLM that is fine-tuned on the CodeLlama family, we then fine-tune CodeLlama-7b-hf for a fair comparison. The evaluation results are shown in Table 9, where we can observe that the fine-tuned results of EFFI-CODE are more efficient and effective compared to those of PIE. For example, the pass@1 of PIE only achieves 19.5% while EFFI-CODE achieves a 37.8% pass@1. In addition, we can observe that EFFI-CODE decreases the ET from 0.42 (s) to 0.39 (s), while PIE reduces the average ET from 0.42 (s) to 0.41 (s).

## 5 CONCLUSION

In this paper, our research addresses a critical gap in the efficiency of code generated by LLMs by introducing the EFFI-CODE dataset, designed to enhance both the correctness and execution efficiency of LLM-generated code via fine-tuning (e.g., SFT, DPO, and ORPO). Through meticulous aggregation, preprocessing, and iterative optimization, we provide a robust resource that significantly boosts the performance of open-source LLMs like DeepSeek-Coder and Qwen. Our experiments reveal substantial improvements, with notable increases in pass rates and decreases in execution time, underscoring the potential of EFFI-CODE to advance the state of code generation in resource-constrained environments. By open-sourcing our model weights, training data, and source code, we aim to foster further research and innovation in this vital area of AI development tools.

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

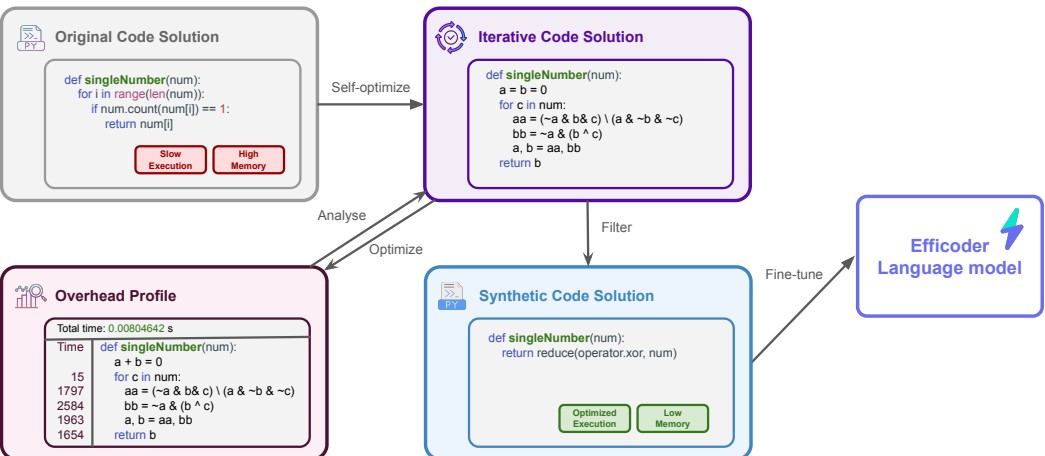

Figure 2: Overview of the construction pipeline for EFFI-CODE.

# A APPENDIX

## A.1 CONSTRUCTION PIPELINE

Figure 2 illustrates the overall framework of EFFI-CODE. We begin by filtering illegal. tasks and collect the initial EFFI-CODE from different open-source datasets. Starting with the original code, we apply self-optimization to enhance efficiency, using test cases to profile execution overhead, and self-improve the code based on the profile. Finally, tasks that fail to have efficiency improvements are removed. We then have our final fine-tuning dataset, EFFI-CODE, which consists of optimized code and rich metadata, designed to train models for generating both efficient and correct code.

## A.2 ADDITIONAL RELATED WORK

**LLMs for Code** The increasing popularity of LLMs for code generation has coincided with the growing availability of open-source code repositories and the need to boost developer productivity. Initial efforts focused on training models specifically for coding tasks, such as CodeT5 (Wang et al., 2021), AlphaCode (Li et al., 2022), CodeGen (Nijkamp et al., 2023), InCoder (Fried et al., 2023), StarCoder (Li et al., 2023a), SantaCoder (Allal et al., 2023), and DeepSeek-Coder (DeepSeekAI, 2023). Contrastingly, models such as Codex (Chen et al., 2021) and CodeLlama (Rozière et al., 2023) represent a subsequent stride, being fine-tuned from foundation models (Brown et al., 2020; Touvron et al., 2023). These code LLMs have been applied to various tasks, including code generation (Chen et al., 2021; Dai et al., 2024), program repair (Haque et al., 2022; Jiang et al., 2023a), automated testing (Lemieux et al., 2023; Deng et al., 2023), code translation (Rozière et al., 2020; Ahmad et al., 2023), type prediction (Mir et al., 2022; Wei et al., 2023), and code summarization (Hasan et al., 2021; Ahmed & Devanbu, 2022). While LLMs have achieved impressive results in code generation tasks like HumanEval (Chen et al., 2021) and MBPP (Austin et al., 2021), their efficiency has received less attention. Recent studies (Shi et al., 2024; Huang et al., 2024b; Niu et al., 2024) have shown that LLM-generated code exhibits lower efficiency in terms of execution time and memory usage compared to canonical solutions. These findings highlight the need for further research and development to improve the efficiency of LLM-generated code. In this work, we propose the first fine-tuning method that significantly improves both the efficiency and correctness of code generated by various LLMs.

## A.3 PROMPT TEMPLATE

> Please continue to complete the function. You are not allowed to modify the given code and do the completion only. Please return all completed functions in a code block. Here is the given code to complete:
> ```python
> {{Prompt}}
> ```

## A.4 EFFICIENCY METRICS

**Execution Time (ET)**   Execution time (ET) measures the average time taken for code execution. Mathematically, ET is defined as:

$$ET = \frac{1}{N} \sum^{N} T_{\text{code}}$$

where $ET$ is the execution time metric, $T_{\text{code}}$ is the execution time of the code (with all the test cases), and $N$ is the number of codes generated by code generation models used for evaluation.

**Normalized Execution Time (NET)**   Normalized Execution Time (NET)[4] measures the execution time required by generated code relative to that of a canonical solution. We define NET as:

$$NET = \frac{1}{N} \sum^{N} \frac{T_{\text{code}}}{T_{\text{canonical}}}$$

where $T_{\text{code}}$ is the execution time of the generated code and $T_{\text{canonical}}$ is the execution time of the canonical solution. A NET value greater than 1 indicates that the generated code is slower than the canonical solution, while a value less than 1 suggests the generated code is faster.

**Max Memory Usage (MU)**   Max Memory Usage (MU) measures the average max memory consumption during code execution. Mathematically, MU is defined as:

$$MU = \frac{1}{N} \sum^{N} M_{\text{code}}$$

where $MU$ is the memory usage metric, $M_{\text{code}}$ is the max memory consumption of the generated code among all the test cases, and $N$ is the number of code instances generated by code generation models used for evaluation. This metric is critical to assess the resource efficiency of generated code, particularly in environments with limited maximum memory capacity.

**Normalized Max Memory Usage (NMU)**   Normalized Max Memory Usage (NMU) quantifies how the max memory efficiency of the generated code compares to the canonical solution. We define NMU as:

$$NMU = \frac{1}{N} \sum^{N} \frac{M_{\text{code}}}{M_{\text{canonical}}}$$

where $NMU$ is the normalized max memory usage metric, $M_{\text{code}}$ is the max memory usage of the generated code, and $M_{\text{canonical}}$ is the max memory usage of the canonical solution. An NMU value less than 1 indicates that the generated code is more memory-efficient than the canonical solution, whereas a value greater than 1 suggests it is less efficient in terms of memory usage. This metric provides a relative measure of the memory optimization in the generated code in comparison to a standard baseline.

---

[4]To demonstrate code-level efficiency, we evaluate the normalized efficiency metrics at the task level, rather than total LLM-generated code / total canonical solutions. For the second calculation strategy, we also provide the scripts in our Github Repo.

**Total Memory Usage (TMU)**   Total Memory Usage (TMU) assesses the efficiency of memory usage throughout the execution of code, taking into account both the magnitude and duration of memory utilization. To calculate TMU, first, monitor and record the memory usage at discrete time intervals during the execution, resulting in a memory usage profile $M(t)$, where $t$ represents time. Then, compute the area under the curve of $M(t)$ over the total execution time, $T_{\text{total}}$, using numerical integration methods such as the trapezoidal rule:

$$TMU = \frac{1}{N} \sum^{N} \int_{0}^{T_{\text{total}}} M(t)\, dt$$

A lower TMU value indicates higher memory efficiency, reflecting an optimized balance between the amount of memory used and the duration of its usage.

**Normalized Total Memory Usage (NTMU)**   The Normalized Total Memory Usage (NTMU) offers a comparison of the dynamic memory efficiency between the generated code and the canonical solution. To determine NTMU, calculate the TMU for both the generated code and the canonical solution. Normalize the TMU of the generated code by dividing it by the TMU of the canonical solution:

$$NTMU = \frac{1}{N} \sum^{N} \frac{TMU_{\text{code}}}{TMU_{\text{canonical}}}$$

where $TMU_{\text{code}}$ is the TMU of the generated code and $TMU_{\text{canonical}}$ is the TMU of the canonical solution. An NTMU value less than 1 signifies that the generated code manages dynamic memory more efficiently compared to the canonical solution, while a value greater than 1 indicates less efficient management of dynamic memory. This metric provides insight into the relative use of dynamic memory of generated code compared to an established benchmark.

## A.5   Additional Related Work

**Learning From Feedback**   A prevalent strategy for improving the behavior of LLMs is learning from feedback, mirroring human learning where individuals refine their actions through trial, error, and correction (Boyd & Fales, 1983; Metcalfe, 2017). Early efforts involve using human feedback to evaluate and refine models (Kreutzer et al., 2018; Ouyang et al., 2022; Glaese et al., 2022). To minimize human intervention, another strategy focuses on automated feedback. These methods iteratively learn from automatically generated feedback signals, understanding the consequences of their actions and adapting their behaviors. The source of this automated feedback can be diverse, ranging from the LLM itself (Madaan et al., 2023; Shinn et al., 2023), external tools (Gou et al., 2023; Lu et al., 2024b) or verifiers (Lu et al., 2024a), external knowledge sources (Gao et al., 2023; Yu et al., 2023) and even generation logits (Yao et al., 2024). In code generation, the program executor is frequently used as a source of feedback for refining the model's initial code. For example, Self-Edit (Zhang et al., 2023a) and Self-Evolve (Jiang et al., 2023b) execute the initial program on example test cases and provide the execution results as feedback, prompting the LLM to refine the code. Self-Debug (Chen et al., 2023) explores using program explanation, unit tests, and program interpreters as feedback types. ALGO (Zhang et al., 2023b) employs a more fine-grained approach by generating a reference oracle program that solves the problem with an exhaustive search. Feedback is then collected by comparing the generated outputs with the oracle. While existing work primarily focuses on using feedback to edit the initial code to ensure correctness, our method explores using overhead profiles to improve the efficiency of the code.

## A.6   What percentage of solutions improved in efficiency but showed degraded correctness

To address this, we provide the evaluation results of degraded correctness (tasks that were correct in the original LLM but became incorrect in the Effi-Code fine-tuned LLM) and upgraded correctness (tasks that were incorrect in the original LLM but became correct in the Effi-Code fine-tuned LLM) in Rebuttal Table 3. We can observe that for all LLMs in the two evaluation datasets, the first scenario, i.e., degraded correctness, is very low. For example, in DeepSeek-Coder-6.7B-base, no tasks went from correct to incorrect after the Effi-Code fine-tuning. However, we can also observe that a large number of incorrect tasks in the original LLMs were correctly addressed by the Effi-Code fine-tuned

Table 10: Evaluation results of degraded and upgraded correctness after Effi-Code fine-tuning on various LLMs.

| Model | Correct → Incorrect | Incorrect → Correct |
|---|---|---|
| **HumanEval** | | |
| DeepSeek-Coder-6.7B-base | 0% | 52.5% |
| DeepSeek-Coder-6.7B-instruct | 4.3% | 37.8% |
| Qwen-Coder-7B | 7.3% | 23.8% |
| Qwen-Coder-7B-instruct | 3.1% | 33.6% |
| **EffiBench** | | |
| DeepSeek-Coder-6.7B-base | 1.2% | 50.3% |
| DeepSeek-Coder-6.7B-instruct | 0.3% | 50.6% |
| Qwen-Coder-7B | 5.9% | 19.7% |
| Qwen-Coder-7B-instruct | 0.1% | 57.8% |

LLMs. For instance, in DeepSeek-Coder-6.7B-base, an additional 52.5% of tasks were addressed by the fine-tuned version.

**Case Study**  To illustrate how the source code generated by EFFI-CODE fine-tuned LLM is more efficient than the source code generated by the LLM without fine-tuning on EFFI-CODE, we provide an example in Figure 3. We can observe that the code generated by Qwen2.5-Coder-7B requires 9.89 (s) to execute all unit tests, while the code generated by EFFI-CODE fine-tuned Qwen2.5-Coder-7B with SFT only requires 0.14 (s) to execute. The key reason is that the code generated by Qwen2.5-Coder-7B requires significantly more recursive calls, as it lacks optimized pruning strategies such as breaking early in redundant paths. This inefficiency leads to a much larger number of computations, ultimately resulting in the observed longer execution time. The code generated by EFFI-CODE fine-tuned Qwen2.5-Coder-7B, on the other hand, incorporates smart optimizations, such as terminating recursion early when certain conditions are met, thereby reducing the overall time complexity.

A.7    CASE EXAMPLE IN IMPROVING EFFICIENCY OF CODE WITH SOAP

We have provided a case example in Figure 4 to demonstrate how SOAP's iterative refinement improves the quality of the solutions. In this example, the initial code generated by DeepSeek-Coder-V2-Lite calculates the Levenshtein distance using a recursive approach, which has an exponential time complexity of $O(3^{(m+n)})$. For longer strings, this recursive method becomes highly inefficient due to the large number of function calls. To optimize the code, the refined version employs dynamic programming, which avoids redundant calculations by filling a distance matrix to compute the Levenshtein distance. The time complexity of the dynamic programming approach is O(mn), where m and n are the lengths of strings a and b, respectively. By filling the distance matrix in a single traversal, the optimized code eliminates redundant calculations, resulting in improved efficiency. The dynamic programming solution leverages the characteristics of optimal substructure and overlapping subproblems, decomposing the problem into smaller subproblems and storing intermediate results to avoid redundant calculations, thereby improving the efficiency of the algorithm. In the provided example, the initial recursive code takes 1.100s to execute, while the optimized dynamic programming code completes execution in just 0.00012s, demonstrating a significant improvement in execution time.

A.8    ROBUSTNESS OF OVERHEAD RESULTS

The overhead results would be affected by the local environments, which causes that the results of Effi-Code fine-tuned LLMs may not able to represent the results of the efficiency profiling in different environments. To address this issue, we have conducted additional experiments and provided more robust evaluation results.

Firstly, we have evaluated the effectiveness of Effi-Code on seven different software-hardware setups, as shown in Rebuttal Table 2. The results demonstrate that Effi-Code fine-tuned LLMs achieve higher

**Task Description**

**Problem:** You are given an integer array `cookies`, where `cookies[i]` denotes the number of cookies in the `ith` bag. You are also given an integer k that denotes the number of children to distribute `all` the bags of cookies to. All the cookies in the same bag must go to the same child and cannot be split up. The `unfairness` of a distribution is defined as the `maximum total` cookies obtained by a single child in the distribution. Return the `minimum` unfairness of all distributions.
solution = Solution() assert solution.distributeCookies([8, 15, 10, 20, 8], 2) == 31 assert solution.distributeCookies([6, 1, 3, 2, 2, 4, 1, 2], 3) == 7

```
solution = Solution()
assert solution.distributeCookies([8, 15, 10, 20, 8], 2) == 31
assert solution.distributeCookies([6, 1, 3, 2, 2, 4, 1, 2], 3) == 7
```

**Qwen2.5-Coder-7B**

```
Execution Time: 9.89 (s)
Max Memory Peak: 24.39 (MB)
Memory Usage: 240.47 (MB*s)
class Solution:
    def distributeCookies(self, cookies,
    ↪   k):
        min_unfairness = float('inf')
        distribution = [0] * k
        def distribute(i):
            nonlocal min_unfairness
            if i == len(cookies):
                min_unfairness =
                ↪   min(min_unfairness,\
                max(distribution))
                return
            for j in range(k):
                distribution[j] +=
                ↪   cookies[i]
                distribute(i + 1)
                distribution[j] -=
                ↪   cookies[i]
        distribute(0)
        return min_unfairness
```

**Qwen2.5-Coder-7B SFT with EFFI-CODE**

```
Execution Time: 0.14 (s)
Max Memory Peak: 24.39 (MB)
Memory Usage: 2.47 (MB*s)
class Solution:
    def distributeCookies(self, cookies,
    ↪   k):
        def backtrack(i):
            nonlocal ans
            if i == len(cookies):
                ans = min(ans,
                ↪   max(children))
                return
            for j in range(k):
                children[j] += cookies[i]
                backtrack(i + 1)
                children[j] -= cookies[i]
                if children[j] == 0:
                    break
        children = [0] * k
        ans = float('inf')
        backtrack(0)
        return ans
```

Figure 3: A case illustration for the task with code generated by Qwen2.5-Coder-7B and EFFI-CODE fine-tuned Qwen2.5-Coder-7B in EffiBench `problem_idx=2305`.

**Task Description**

**Problem: >** In information theory and computer science, the Levenshtein distance is a string metric for measuring the difference between two sequences. Informally, the Levenshtein distance between two words is the minimum number of single-character edits (i.e. insertions, deletions or substitutions) required to change one word into the other.
(http://en.wikipedia.org/wiki/Levenshtein_distance)
Your task is to implement a function which calculates the Levenshtein distance for two arbitrary strings.

**Optimized**

```python
# Execution Time: 0.00012s
def levenshtein(a: str, b: str) -> int:
    distance_matrix = [[0] * (len(b) + 1)\
     for _ in range(len(a) + 1)]
    for i in range(len(a) + 1):
        distance_matrix[i][0] = i
    for j in range(len(b) + 1):
        distance_matrix[0][j] = j
    for i in range(1, len(a) + 1):
        for j in range(1, len(b) + 1):
            if a[i-1] == b[j-1]:
                distance_matrix[i][j] = \
                distance_matrix[i-1][j-1]
            else:
                distance_matrix[i][j] =
                ↪  min(
                distance_matrix[i-1][j] +
                ↪  1,
                distance_matrix[i][j-1] +
                ↪  1,
                distance_matrix[i-1][j-1] +
                ↪  1)
    return distance_matrix[len(a)][len(b)]
```

**Initial**

```python
# Execution Time: 1.100s
def levenshtein(a, b):
    if not a:
        return len(b)
    if not b:
        return len(a)
    if a[0] == b[0]:
        return levenshtein(a[1:], b[1:])
    return 1 + min(levenshtein(a[1:], b),
                levenshtein(a, b[1:]),
                levenshtein(a[1:], b[1:]))
```

Figure 4: A case for the task with code and SOAP refined version. The lower left panel shows the initial completion generated by an LLM, its profile shows its inefficiency, which requires 1.100s to execute. The lower right panel shows the final efficient answer output by applying SOAP, which only requires 0.00012s to execute.

| Setup | ET | NET | MU | NMU | TMU | NTMU |
|---|---|---|---|---|---|---|
| Python 3.11.10 - Intel(R) Xeon(R) Platinum 8336C CPU @ 2.30GHz | | | | | | |
| Qwen2.5-Coder-7B | 0.59 | 1.95 | 61.95 | 0.99 | 24.29 | 1.83 |
| +Effi-Code | 0.40 | 1.01 | 61.96 | 0.99 | 18.74 | 1.02 |
| Python 3.11.10 - Intel(R) Xeon(R) Silver 4216 CPU @ 2.10GHz | | | | | | |
| Qwen2.5-Coder-7B | 0.28 | 1.63 | 36.15 | 1.00 | 20.01 | 1.88 |
| + SFT | 0.25 | 1.38 | 36.52 | 1.01 | 19.85 | 1.56 |
| Python 3.11.10 - Intel(R) Xeon(R) Silver 4116 CPU @ 2.10GHz | | | | | | |
| Qwen2.5-Coder-7B | 0.35 | 1.45 | 36.14 | 1.00 | 24.28 | 1.63 |
| + SFT | 0.22 | 1.01 | 36.51 | 1.01 | 15.26 | 1.09 |
| Python 3.11.4 - Intel(R) Xeon(R) Silver 4216 CPU @ 2.10GHz | | | | | | |
| Qwen2.5-Coder-7B | 0.67 | 1.16 | 61.43 | 1.00 | 40.01 | 1.22 |
| +Effi-Code | 0.58 | 1.02 | 60.77 | 0.97 | 32.50 | 1.03 |
| Python 3.11.0 - Intel(R) Xeon(R) Silver 4216 CPU @ 2.10GHz | | | | | | |
| Qwen2.5-Coder-7B | 0.28 | 1.64 | 34.55 | 1.00 | 19.39 | 1.87 |
| + SFT | 0.25 | 1.39 | 34.90 | 1.02 | 20.03 | 1.59 |
| Python 3.9.0 - Intel(R) Xeon(R) Silver 4216 CPU @ 2.10GHz | | | | | | |
| Qwen2.5-Coder-7B | 0.30 | 1.60 | 34.26 | 1.01 | 21.02 | 2.10 |
| +Effi-Code | 0.24 | 1.20 | 34.52 | 1.02 | 19.84 | 1.32 |
| Python 3.10.0 - Intel(R) Xeon(R) Silver 4216 CPU @ 2.10GHz | | | | | | |
| Qwen2.5-Coder-7B | 0.29 | 1.63 | 33.26 | 1.01 | 20.32 | 2.16 |
| + SFT | 0.26 | 1.43 | 33.50 | 1.02 | 19.53 | 1.61 |

Table 11: Rebuttal Table 2: Evaluation results of Effi-Code's effectiveness on different software-hardware setups.

efficiency than the original LLMs across all setups. For example, in the environment of Python 3.11.10 - Intel(R) Xeon(R) Platinum 8336C CPU @ 2.30GHz, the average execution time decreases from 0.59s to 0.40s when using Effi-Code to fine-tune Qwen2.5-Coder-7B, reducing the average execution time by 32%.

Secondly, we clarify that for the same setup, where we evaluate the efficiency of LLM-generated code several times, the efficiency results are consistent. As shown in Paper Table 8, where we execute the LLM-generated code five times, the standard deviation of execution time (ET) is 0.00548 (s), indicating that the evaluation results are consistent and reliable for a given setup.

Finally, our evaluation setup follows the practices established in recent works on benchmarking the efficiency of automatically generated code, such as Mercury Du et al. (2024), Effibench Huang et al. (2024b), and SOAP Huang et al. (2024a). By adhering to these benchmarks, we ensure that our evaluation is in line with the current standards in the field.

## A.9 ADDITIONAL EFFI-CODE INSTRUCT TUNING LLMS

We have conducted additional experiments by fine-tuning Effi-Code on five more open-source LLMs. We have carefully selected these LLMs based on their popularity and performance in code generation tasks. The results are presented in Table 12, demonstrating the effectiveness of Effi-Code in improving the efficiency of the generated code across various LLMs. We can observe that all the evaluated LLMs exhibit improvements in both code efficiency and pass@1 metrics after fine-tuning with Effi-Code. For instance, CodeLlama-13B-hf shows a significant reduction in execution time (ET) from 0.86s to 0.13s on average for correctly overlapped tasks, which reduces execution time by 84.88%. In addition, we can also observe that the pass@1 of CodeLlama-13B-hf generated code increases from 7.9% to 28.8%, which also increases pass@1 by 20.9% compared to the original LLM. These

| Model | ET | NET | MU | NMU | TMU | NTMU | Overlap | pass@1 |
|---|---|---|---|---|---|---|---|---|
| starcoder2-7b | 0.41 | 2.22 | 77.86 | 1.62 | 215.26 | 23.33 | 16.4 | 23.6 |
| + SFT | 0.40 | 2.21 | 36.58 | 1.00 | 14.63 | 3.83 | 16.4 | 28.8 |
| starcoder2-15b | 0.29 | 1.52 | 41.28 | 1.00 | 34.09 | 1.85 | 17.9 | 21.2 |
| + SFT | 0.20 | 1.08 | 42.18 | 1.04 | 20.07 | 1.06 | 17.9 | 42.8 |
| CodeLlama-13b-hf | 0.86 | 6.57 | 34.32 | 1.12 | 55.69 | 11.02 | 5.3 | 7.9 |
| + SFT | 0.13 | 0.97 | 31.02 | 1.00 | 3.71 | 0.98 | 5.3 | 28.8 |
| codegemma-7b | 0.11 | 0.95 | 26.25 | 1.00 | 1.62 | 0.95 | 0.2 | 0.2 |
| + SFT | 0.10 | 0.94 | 26.01 | 0.98 | 1.46 | 0.89 | 0.2 | 35.1 |
| DeepSeek-Coder-6.7b-base | 0.44 | 2.61 | 57.24 | 1.26 | 54.57 | 7.94 | 7.3 | 8.5 |
| + SFT (Ours) | 0.29 | 2.08 | 50.58 | 1.00 | 17.25 | 2.79 | 7.3 | 57.6 |
| DeepSeek-Coder-6.7b-instruct | 0.14 | 1.00 | 38.36 | 1.00 | 4.21 | 0.97 | 1.0 | 1.3 |
| + SFT (Ours) | 0.13 | 0.93 | 38.31 | 1.00 | 4.01 | 0.92 | 1.0 | 51.6 |
| Qwen2.5-Coder-7B | 0.26 | 1.79 | 38.06 | 1.01 | 18.30 | 2.74 | 44.2 | 50.1 |
| + SFT (Ours) | 0.21 | 1.45 | 38.15 | 1.01 | 15.88 | 1.70 | 44.2 | 63.9 |
| Qwen2.5-Coder-7B-Instruct | 0.44 | 3.96 | 28.62 | 1.00 | 10.17 | 5.43 | 3.2 | 3.3 |
| + SFT (Ours) | 0.43 | 3.88 | 28.59 | 1.00 | 10.10 | 5.37 | 3.2 | 61.0 |

Table 12: Comparison of Effi-Code across different open-source LLMs.

Table 13: Efficiency results on the HumanEval-X (C++) dataset.

| HumanEval-X (C++) | ET (s) | NET | MU (KB) | NMU | TMU (KB*s) | NTMU |
|---|---|---|---|---|---|---|
| DeepSeek-Coder-6.7B-base | 0.44 | 1.4 | 83.9 | 1.3 | 25.2 | 1.9 |
| SFT with Effi-Code | 0.32 | 1.0 | 71.3 | 1.1 | 18.9 | 1.4 |

additional experiments on a diverse set of open-source LLMs further validate the generalizability and effectiveness of our proposed Effi-Code dataset.

## A.10 EXPERIMENTAL RESULTS ON HUMANEVAL-X (C++) DATASET

We have conducted additional experiments on the HumanEval-X (C++) dataset and provided the efficiency results in Table 13. We can observe that the efficiency of LLM-generated code also improved with Effi-Code fine-tuned LLM. For instance, the average execution time (ET) for the overlapped code decreases from 0.44s to 0.32s, resulting in a 27% reduction in execution time.

Furthermore, to investigate whether the efficiency of the code generated by Effi-Code fine-tuned LLMs can be further enhanced once we add additional efficient C++ code into the Effi-Code dataset, we have followed the pipeline of Effi-Code and constructed an Effi-Code (C++) subset containing 3,322 C++ tasks. We then fine-tuned LLMs using three different setups: Effi-Code (Py), Effi-Code (C++), and Effi-Code (C++) + Effi-Code (Py). The evaluation results, presented in Table 14, reveal several interesting findings.

Firstly, LLMs fine-tuned on the Effi-Code datasets generate more efficient code compared to the original LLM-generated code. For example, the average execution time for Qwen2.5-Coder-7B generated code is 0.35s, while the Effi-Code (Py) fine-tuned LLMs require only 0.17s on average for overlapped tasks, resulting in a 51.4% reduction in average execution time.

Secondly, when we utilize Effi-Code (C++) and Effi-Code (Py) + Effi-Code (C++) to fine-tune LLMs, the overhead of LLM-generated code is further decreased. The average execution time for overlapped code decreases from 0.17s to 0.16s, and the memory peak (MU) also decreases from 46.71MB to 43.72MB. These results indicate that by incorporating C++ source code to guide LLM fine-tuning, LLMs may learn additional optimization strategies.

Table 14: Efficiency results on the EffiBench dataset with different fine-tuning setups.

| EffiBench | ET (s) | NET | MU (MB) | NMU | TMU (MB*s) | NTMU |
|---|---|---|---|---|---|---|
| Qwen2.5-Coder-7B | 0.35 | 2.01 | 43.72 | 0.99 | 12.35 | 0.98 |
| EffiCode (Py) | 0.17 | 1.02 | 46.71 | 1.12 | 7.53 | 1.29 |
| EffiCode (CPP) | 0.17 | 1.01 | 43.74 | 0.99 | 6.65 | 1.04 |
| EffiCode (Py) + EffiCode (CPP) | 0.16 | 1.00 | 43.72 | 0.99 | 6.01 | 0.99 |

Table 15: Efficiency results on the EffiBench dataset with different fine-tuning setups.

| EffiBench | ET (s) | NET | MU (MB) | NMU | TMU (MB*s) | NTMU |
|---|---|---|---|---|---|---|
| Qwen2.5-Coder-7B | 0.49 | 3.50 | 25.69 | 1.00 | 10.75 | 4.78 |
| +Effi-Code + non-algorithmic | 0.19 | 1.16 | 25.67 | 1.00 | 4.17 | 1.17 |
| +Effi-Code | 0.19 | 1.15 | 25.69 | 1.00 | 4.07 | 1.15 |

A.11    INCORPORATING NON-ALGORITHMIC TASKS

We have conducted additional experiments and provided the evaluation results in Table 15, which compares the performance of the original Qwen2.5-Coder-7B, the model fine-tuned on Effi-Code, and the model fine-tuned on Effi-Code + non-algorithmic tasks (optimized).

As shown in Table 15, when we fine-tune Qwen2.5-Coder-7B on either Effi-Code or Effi-Code + non-algorithmic tasks, the efficiency of LLM-generated code improves. For instance, the average execution time for overlapped correct tasks decreases from 0.49s to 0.19s for both Effi-Code and Effi-Code + non-algorithmic tasks fine-tuned Qwen2.5-Coder-7B.

However, we also observe that the TMU of the Effi-Code fine-tuned Qwen2.5-Coder-7B is lower than the model fine-tuned on Effi-Code + non-algorithmic tasks. Specifically, the Effi-Code + non-algorithmic tasks fine-tuned Qwen2.5-Coder-7B decreases the average TMU for overlapped correct code from 10.75 MB*s to 4.17 MB*s. In contrast, Qwen2.5-Coder-7B fine-tuned only on Effi-Code further reduces the TMU from 4.17 MB*s to 4.07 MB*s.

Our results indicate that while incorporating non-algorithmic tasks in the fine-tuning process can lead to improvements in code efficiency, focusing solely on algorithmic tasks, as done in Effi-Code, may yield even better results. Nonetheless, we acknowledge the potential benefits of broadening the scope to include non-algorithmic optimizations, as it can enhance the real-world implications of Effi-Code. In future work, we plan to explore the integration of non-algorithmic tasks more comprehensively while maintaining the focus on algorithmic optimization.

A.12    EFFICIENCY RESULTS OF PIE AND EFFI-CODE FINE-TUNED LLM IN PIE TEST SET

We also provided the efficiency results of the PIE fine-tuned CodeLlama, and Effi-Code fine-tuned CodeLlama in Table 16. For each task, we requested each LLM to generate efficient code. The results demonstrate that for the PIE test set, the efficiency of the code generated by the Effi-Code fine-tuned CodeLlama-7B is also better than that of the PIE fine-tuned CodeLlama-7B. Specifically, the average execution time for overlapping correct code generated by the PIE fine-tuned LLM is 0.39s. However, the Effi-Code fine-tuned CodeLlama further reduces this average execution time from 0.39s to 0.34s, resulting in an additional 8% reduction in execution time.

Table 16: Efficiency comparison of CodeLlama-7B fine-tuned on PIE and Effi-Code, evaluated on the PIE test set.

| PIE Test Set | ET (s) | NET | MU (MB) | NMU | TMU (MB*s) | NTMU |
|---|---|---|---|---|---|---|
| CodeLlama7B+PIE | 0.39 | 0.84 | 7.3 | 0.93 | 1.7 | 0.95 |
| CodeLlama7B+Effi-Code | 0.34 | 0.76 | 7.2 | 0.91 | 1.5 | 0.88 |

Table 17: Efficiency comparison of different methods on the HumanEval dataset.

| Method | ET | NET | MU | NMU | TMU | NTMU | overlapped | pass@1 |
|---|---|---|---|---|---|---|---|---|
| CodeLlama-7B-hf | 1.40 | 1.02 | 62.36 | 0.99 | 63.49 | 0.98 | 1.2 | 12.2 |
| Supersonic | 1.24 | 0.90 | 63.39 | 1.01 | 63.18 | 0.98 | 1.2 | 15.2 |
| PIE | 1.32 | 0.96 | 63.24 | 1.00 | 65.28 | 1.03 | 1.2 | 19.5 |
| Effi-Code | 1.21 | 0.87 | 62.06 | 0.99 | 56.05 | 0.87 | 1.2 | 37.8 |
| DeepSeek-Coder-6.7B-base | 2.30 | 1.00 | 75.35 | 1.00 | 166.68 | 0.97 | 4.9 | 7.3 |
| Mercury | 2.29 | 0.99 | 75.30 | 1.00 | 174.05 | 0.99 | 4.9 | 29.9 |
| Effi-Code | 2.24 | 0.94 | 75.30 | 1.00 | 160.10 | 0.92 | 4.9 | 51.8 |

## A.13 EVALUATION RESULTS WITH ADDITIONAL BASELINES

We provide the evaluation results of Supersonic, PIE, Mercury, and Effi-Code in Table 17. We currently only have the inference results of Mercury in the DeepSeek-Coder-6.7B-base, so we compare the efficiency of Mercury and Effi-Code in the DeepSeek-Coder-6.7B-base. For Supersonic and PIE, we compare the efficiency results in CodeLlama-7B-hf. Furthermore, as the training set of Mercury contains some tasks in EffiBench, for a fair comparison, we evaluate the efficiency results in the HumanEval dataset.

As shown in Table 17, we can observe that for both models, Effi-Code achieves state-of-the-art (SOTA) performance compared to the baselines. For example, in CodeLlama-7B-hf, the average execution time for Supersonic decreases from 1.40s to 1.24s on average for all overlapping correct tasks, while Effi-Code further decreases the average execution time from 1.24s to 1.21s. Compared to the solution generated by CodeLlama-7B-hf, the average execution time was reduced by 16.7%.

