# OpenReview forum: "Effi-Code: Unleashing Code Efficiency in Language Models"
_ICLR.cc/2025/Conference — Submitted to ICLR 2025_

### Official Review · Reviewer_HPnS · 2024-10-24

**Soundness:** 1
**Presentation:** 2
**Contribution:** 1
**Rating:** 1
**Confidence:** 5

**Summary:**

This paper introduces a new benchmark (Effi-Code) for fine-tuning existing LLMs to generate more efficient code. More specifically, the authors collect code snippets from existing code generation tasks and apply multiple pre-processing steps. These steps include filtering code snippets with risky operations, generating test cases for validating generated code, etc. Moreover, the authors employ Self-Optimization based on overheAd Profile (SOAP) for optimizing the collected code snippets for efficiency. The code snippets are executed against generated tests to collect the execution time and memory usage of code snippets. The profiling results are then provided to a judge LLM (DeepSeek-Coder) along with source code and description to construct an efficient version of the code. The generated efficient code snippets are then used to fine-tune open-source LLMs (deepseek-coder, qwen-code). LLMs fine-tuned on efficient code snippets significantly outperforms the base version of themselves both in terms of efficiency (e.g., execution time, memory usage) and correctness (e.g., pass@1) on HumanEval and Effibench datasets. The authors further perform certain ablation studies to understand other research questions.

**Strengths:**

- Studying the efficiency of code generated by LLM is an important problem. The authors propose a technique to generate efficient code snippets which can be used for fine-tuning existing LLMs.
- Empirical results performed by the authors depict significant increase in performance of LLMs, both in terms of efficiency and correctness
- A comparison study is performed to compare the performance of Effi-Code with PIE.

**Weaknesses:**

False claims / Repititive work:
- In the abstract, the authors falsely claim "We introduce a Self-Optimization process based on Overhead Profiling", however, this contribution is already made in a published paper (Soap: Enhancing efficiency of generated code via self-optimization. arXiv preprint arXiv:2405.15189).

- The test case generation technique used in this paper is already made in a published paper (Effibench: Benchmarking the efficiency
of automatically generated code. arXiv preprint arXiv:2402.02037). There is not enough novelty for test case generation in this work.

Too much reliance on LLM:
- The authors decided to fully rely on LLMs when filtering code snippets for risky operations and test case generation. However, in the case of filtering, a simple static analysis approach (without executing the code) could have been used given that static analysis is more sound than using LLMs.
- As for the test case generation, I am wondering why didn't the authors consider using off-the-shelf tools like CODAMOSA (https://dl.acm.org/doi/10.1109/icse48619.2023.00085) and other SOTA test generators from SE community. Again, relying only on GPT-3.5 is not sufficient in generating proper test cases.
- There is no background work on the use of LLM as a judge. I don't see any soundness in using DeepSeek-Coder to construct efficient code snippets.

Presentation issues:
- Some parts of the paper is not well-written. For instance please see lines 73-76.

There are some descrepancies in the paper as well:
- You mention in the instruction that you collected data from 8 sources, however, in section 3 it says seven sources.

**Questions:**

- Why should somebody trust the filtering for risky operations done by the LLM? Is there any manual investigation of what has been flagged by the LLM as risky operation? Are there any False Positives or False Negatives?
- What is the novelty in test case generation component of Effi-Code? Is it any different from Effibench?
- How do the authors support the use of LLM as a judge for constructing efficient solutions?
- What is the quality of generated test cases by GPT-3.5? Specifically, is it possible to get the code coverage, branch coverage, etc.?
- How do the authors filter non-algorithmic tasks.
- Have the authors performed a deep investigation on why the pass@1 scores of the model increases significantly? For instance, the pass@1 value of deepseek-coder-6.7b-base goes from 7.3 to 59.8. What are the chances the "quality" of these code snippets regardless of them being efficient is causing the increase in performance?

---

> ### Author Response · Authors · 2024-11-21
>
> Dear Reviewer HPnS,
>
>
> Thank you for recognizing the importance of studying the efficiency of code generated by LLMs. We are pleased that you appreciate our proposed technique for generating efficient code snippets that can be used for fine-tuning existing LLMs. Your acknowledgment of the significant increase in the performance of LLMs, both in terms of efficiency and correctness, as demonstrated by our empirical results, validates the effectiveness of our approach. To address your concerns regarding the weaknesses and questions you have raised, we will provide a detailed point-by-point response below. We hope that our clarifications, additional experiments, and thorough explanations will help alleviate your concerns and demonstrate the robustness and potential of our approach.
>
> **False claims / Repetitive work**
>
> We would like to clarify that the steps such as Section 3.3 using SOAP [2] to optimize the efficiency of LLM-generated code and Step 3 generating test cases to measure the efficiency of LLM-generated code, are introduced as the detailed component of our framework to construct the Effi-Code dataset. We do not treat these as our key contribution.
>
>
>
> We clarify that the key contributions of our work are as follows:
>
> - We provide a framework to inspire researchers to construct code generation datasets containing efficient solutions for each code generation task (task description), which is versatile and can be adapted to different programming languages and leverage various existing data sources. Unlike some other code generation datasets that rely on powerful models (e.g., GPT-4), our framework can be implemented using more accessible models like DeepSeek-Coder. The framework provides a systematic method for researchers to enhance existing datasets or create new ones focused on code efficiency across different languages and domains.
>
> - Based on our proposed framework, we release the Effi-Code dataset, which is the first instruct-tuning dataset that focuses on improving the efficiency of LLM-generated code. The primary purpose of Effi-Code is to instruct and fine-tune LLMs to ensure that the LLM-generated code for each task is more efficient than the code generated by the original LLM.
>
> - We use Effi-Code to fine-tune several widely used LLMs and will release these models on the Hugging Face website in the final version. Different from existing datasets that are used to finetune the LLMs to improve the pass@1 of LLM-generated code, our evaluation results demonstrate that both the pass@1 and the efficiency results would be improved for LLMs finetuned on our Effi-Code dataset.
>
>
> Each step in the Effi-Code construction framework demonstrates how we process candidate tasks, filter out lower quality and risky tasks, and construct an efficient solution for each task. We introduce these techniques used in our Effi-Code construction framework. However, it is important to note that the individual steps in our construction framework are not considered our primary novelty or contribution; rather, they serve to illustrate our methodology.
>
>
>
> **W1: In the abstract, the authors falsely claim**
>
>
> Thank you for your comment. We apologize for the confusion caused by our wording in the abstract. Our intention was not to claim the Self-Optimization process based on Overhead Profiling (SOAP) as a new method, and take it as our core contribution. We acknowledge that SOAP was proposed in the previously published paper and already accepted by NeurIPS. In our manuscript, we have also cited the aforementioned paper in both the Introduction and Method sections to give proper credit to the original work. We will revise the abstract to clarify that our core contribution is the generation of the Effi-Code dataset using the SOAP method, rather than proposing SOAP itself.
>
>
> Next, in our work, we aim to express that we generate a dataset (Effi-Code dataset) based on the SOAP method, which ensures the efficiency of the generated code is better than the original solution provided by each task in our collected candidate tasks. Similar to SOAP, some methods such as Mercury and PIE can be used to generate more efficient solutions compared to the original solution provided by each task in our collected tasks. In contrast, we use SOAP to optimize the efficiency of the original solution are due to SOAP can generate the most efficient solutions and more correct solutions compared to other methods in our preliminary experiments.
>
> The key contributions of our paper are
>
> - We provide a framework to construct a code generation dataset (Effi-Code) that contains efficient solutions for each task.
> Based on our proposed framework, we release the Effi-Code. To the best of our knowledge, it is the first instruct tuning dataset that focuses on improving the efficiency of LLM-generated code.
> - We reveal that fine-tuning LLMs using Effi-Code (efficient solutions for each task) ensures that LLMs generate more efficient solutions compared to baselines.

---

> > ### Author Response · Authors · 2024-11-21
> >
> > **W2: The test case generation technique used in this paper has already been made in a published paper (Effibench: Benchmarking the efficiency of automatically generated code. arXiv preprint arXiv:2402.02037). There is not enough novelty for test case generation in this work.**
> >
> > Response:
> >
> > First, we clarify that the test case generation process is not our key contribution. We introduce test case generation are due to we provide the detailed framework to optimize the efficiency of the initial code and then construct the Effi-Code dataset.
> >
> > Next, both Effi-Code and EffiBench utilize GPT-3.5-turbo for test case generation, there are notable differences in their approaches, which contribute to the novelty of Effi-Code's technique.
> >
> > 1. **Direct vs. Indirect Generation**:
> >    - **Effi-Code**: Directly requests GPT-3.5-turbo to generate test cases specifically designed to measure code efficiency.
> >    - **EffiBench**: Takes an indirect approach by first requesting GPT-3.5-turbo to generate a test case generator, which is then used to generate the actual test cases.
> >
> > 2. **Purpose and Scale**:
> >    - **Effi-Code**: Focuses on generating targeted test cases solely for the purpose of measuring code efficiency. These test cases are typically smaller in scale and less complex compared to those generated by EffiBench.
> >    - **EffiBench**: Aims to generate large and comprehensive test cases, with an average token length exceeding 2,000. The generated test case generator in EffiBench is designed to produce a wide range of test cases efficiently.
> >
> > 3. **Efficiency and Overhead**:
> >    - **Effi-Code**: Direct generation of efficiency-measuring test cases through GPT-3.5-turbo allows for a streamlined process without the need for an intermediate generator. This approach reduces the overhead associated with generating and managing a separate test case generator.
> >    - **EffiBench**: Its two-step process, involving the generation of a test case generator and its subsequent use to generate test cases, introduces additional complexity and overhead compared to Effi-Code's direct generation approach.
> >
> >
> > **W3: The authors decided to fully rely on LLMs when filtering code snippets for risky operations and test case generation. However, in the case of filtering, a simple static analysis approach (without executing the code) could have been used given that static analysis is more sound than using LLMs.**
> >
> > Response:
> >
> > Thanks for your suggestion! The key reason our experiments do not directly use a static analysis approach such as AST is that the statistical analysis ignores some of the risky operations and then once we execute some code in local environments, the system files may be deleted. For example, we provide a case example below:
> >
> > ```python
> > def calculate_result(expression):
> > return eval(expression)
> >
> > user_input = input("Enter a mathematical expression: ")
> > result = calculate_result(user_input)
> > print("Result:", result)
> > ```
> >
> > In this code snippet, the `eval()` function is used to evaluate the user input, which can execute any valid Python code, including code that can have unintended or malicious consequences. Static analysis tools may not always detect the risk associated with using `eval()` because it depends on the dynamic input provided at runtime. While static analysis can be effective in identifying certain types of risky operations, it has limitations in detecting dynamic and runtime-dependent risks.
> >
> > On the other hand, when we request GPT-3.5-turbo to analyze whether the code may exist in risky operations, where we require GPT-3.5-turbo to deeply analyze whether some trick may be used in the code to attack the local environments, GPT-3.5-turbo can then identify above potential risks that may not be apparent through static analysis alone.
> >
> > In addition to using LLMs for filtering risky operations, we also employ a set of predefined rules before executing the code. These rules involve analyzing the code for specific operations such as `mkdir`, `rm`, and other potentially dangerous functions. By applying these rules, we can further ensure that most of the risky operations are filtered out before execution. However, we acknowledge that relying solely on LLMs and predefined rules for filtering risky operations may not be the most robust approach. If the reviewers have recommendations for methods that are more solid and effective than LLMs in detecting risky operations without executing the code, we would be open to comparing and evaluating those methods.

---

> > > ### Author Response · Authors · 2024-11-21
> > >
> > > **W4: As for the test case generation, I am wondering why didn't the authors consider using off-the-shelf tools like CODAMOSA (https://dl.acm.org/doi/10.1109/icse48619.2023.00085) and other SOTA test generators from SE community. Again, relying only on GPT-3.5-turbo is not sufficient for generating proper test cases.**
> > >
> > >
> > >
> > > Response:
> > >
> > >
> > >
> > > Thank you for your suggestion. The primary reason for not using CODAMOSA and other test case generation methods from the software engineering community, such as MOSA and DynaMOSA, is that these strategies require significant execution time. For instance, CODAMOSA typically needs about 1-10 minutes (default setting) to generate test cases for each test module. Given that the number of tasks in Step 3 requiring test case generation is 78K, using traditional software engineering methods would take several months. However, directly using GPT-3.5-turbo to generate massive test cases and then filtering out incorrect ones based on the canonical solution provided by the dataset can reduce the testing time to less than 1 hour.
> > >
> > >
> > >
> > > To evaluate the effectiveness of CODAMOSA (with a test case generation time of 1 minute per task) and our direct use of GPT-3.5-turbo for test case generation, we randomly selected 100 tasks from the Step 3 candidate tasks to expedite the testing process. The evaluation results are shown in Rebuttal Table 1. We observed that although the code coverage of CODAMOSA is slightly higher than Effi-Code, the execution time of CODAMOSA is considerably longer. For example, CODAMOSA's code line coverage is only 1.08% higher than Effi-Code, but it requires 100 minutes to generate test cases, while Effi-Code only needs 8.48 seconds.
> > >
> > > | Generation Strategy | Line Coverage | Branch Coverage | Time       |
> > > |---------------------|---------------|-----------------|------------|
> > > | CODAMOSA            | 97.81%        | 98.53%          | 100 minutes|
> > > | Effi-Code           | 96.73%        | 97.18%          | 8.48 seconds|
> > >
> > > *Rebuttal Table 1: Comparison of code coverage and execution time between CODAMOSA and our Effi-Code strategy for test case generation across a sample of 100 tasks.*
> > >
> > >
> > > **W5: There is no background work on the use of LLM as a judge. I don't see any soundness in using DeepSeek-Coder to construct efficient code snippets.**
> > >
> > >
> > > Good question! We clarify the detailed information in two parts:
> > >
> > >
> > > *First, how do we ensure that the optimized code snippets are more efficient?*
> > >
> > > The key reason we can obtain more efficient code snippets compared to the original solutions lies in Section 3.3 and Step 6. Specifically, for each task, we first provide the execution time and memory usage profiler to the LLMs and then require the LLMs to optimize the code's efficiency. After that, we analyze whether the efficiency of the LLM-generated code has improved over five iterations. If the efficiency of the newly generated code does not improve, we filter these tasks from our candidate pool. This ensures that for each task in our constructed Effi-Code dataset, the provided solution is more efficient than the original solution.
> > >
> > >
> > > *Second, why do we choose DeepSeek-Coder rather than other open-/closed-source LLMs to optimize the code?*
> > >
> > > To decide which LLM to use for optimizing the code efficiency for each task, we conducted a preliminary study on APPS, analyzing the use of different LLMs. The evaluation results are shown in Rebuttal Table 2, where we use three LLMs: DeepSeek-Coder-V2-Lite-Instruct, Meta-Llama-3-70B-Instruct, and OpenCodeInterpreter-DS-33B to optimize the source code in Step 4 with five iterations. We report the execution time (ET), memory usage/peak (MU), total memory usage (TMU) for the overlapped tasks, overlap percentage, and pass@1 results.
> > >
> > >
> > >
> > > We observe that compared to the other two LLMs, DeepSeek-Coder-V2-Lite-Instruct can generate competitively efficient code similar to Meta-Llama-3-70B-Instruct, and it also has the highest pass@1 for our Step 4 APPS tasks. This ensures that if we use DeepSeek-Coder-V2-Lite-Instruct to optimize the initial code for each task, we can generate a larger number of correct and efficient codes compared to codes with other LLMs. Therefore, in our experiments, we use DeepSeek-Coder-V2-Lite-Instruct to optimize the initial code for all tasks.
> > >
> > >
> > > | Model                           | Model Size | ET   | MU     | TMU   | Overlap | Pass@1 |
> > > |---------------------------------|------------|------|--------|-------|---------|--------|
> > > | DeepSeek-Coder-V2-Lite-Instruct | 16B        | 2.76s| 200.40 | 89.28 | 3.4%    | 56.5%  |
> > > | Meta-Llama-3-70B-Instruct       | 70B        | 2.76s| 200.48 | 89.10 | 3.4%    | 10.7%  |
> > > | OpenCodeInterpreter-DS-33B      | 33B        | 3.52s| 204.57 | 115.56| 3.4%    | 31.2%  |
> > >
> > > *Rebuttal Table 2: Evaluation of three language models on optimizing code efficiency across the APPS dataset.*

---

> > > > ### Author Response · Authors · 2024-11-21
> > > >
> > > > **W6: Some parts of the paper are not well-written. For instance please see lines 73-76.**
> > > >
> > > >
> > > >
> > > > Thanks for your reminder. We have revised our manuscript Lines 77-81 to fix these problems.
> > > >
> > > >
> > > >
> > > >
> > > >
> > > > **W7: You mention in the instruction that you collected data from 8 sources, however, in section 3 it says seven sources.**
> > > >
> > > >
> > > >
> > > > Thanks for your reminder. We have revised our manuscript Line 147 to fix these problems.
> > > >
> > > >
> > > >
> > > > **Q1: Why should somebody trust the filtering for risky operations done by the LLM? Is there any manual investigation of what has been flagged by the LLM as a risky operation? Are there any False Positives or False Negatives?**
> > > >
> > > >
> > > >
> > > > Response:
> > > >
> > > > We would like to clarify that the primary reason for using an LLM to filter risky operations from our collected candidate tasks is to ensure the security of our local execution environments. During our initial analysis, we observed that some tasks contained risky operations that caused system files to be deleted when executing the canonical solution. To protect our local environments, we decided to utilize an LLM to analyze whether any of the collected candidate tasks contained risky operations, such as deleting system files or removing files in the execution folder. However, we want to clarify that trusting the LLM's filtering of risky operations is not strictly necessary. Researchers can choose to run the potentially risky code in a simulated environment, such as PIE, which provides isolated execution environments regardless of whether system files might be modified or deleted.
> > > >
> > > > In addition to using GPT-3.5-turbo for filtering tasks, we also implemented additional rules to filter tasks (i.e., false negatives ignored by GPT-3.5-turbo) when executing the code locally. For example, we analyze the task-provided canonical solution to check for the presence of operations like `mkdir` and `rm`. If these operations are found in the canonical solution, we remove the corresponding tasks from our collected candidate tasks to ensure that the file system remains unmodified during code execution.
> > > >
> > > > To assess the presence of false positives (i.e., tasks that do not involve risky operations but are incorrectly classified by GPT-3.5-turbo as containing them), we randomly selected 20 tasks from our filtered tasks and manually checked whether each task actually contained risky operations. After this manual inspection, we found that all 20 tasks indeed contained risky operations, indicating that there were no false positives in this sample. **It is worth noting that even if false negatives (i.e., tasks with risky operations that were not flagged by GPT-3.5-turbo) exist in the Effi-Code dataset, the filtering process is still valuable as it reduces the need for manual effort in analyzing each task for risky operations, which can be time-consuming and labor-intensive.**
> > > >
> > > >
> > > >
> > > > In summary, while the LLM-based filtering of risky operations may not be perfect, it serves as an effective first line of defense in protecting local execution environments. Researchers can further mitigate risks by using simulated environments or applying additional filtering rules. The manual inspection of a sample of filtered tasks suggests that false positives are rare, and the benefits of automated filtering outweigh the potential drawbacks of false negatives.
> > > >
> > > >
> > > >
> > > > **Q2: What is the novelty in the test case generation component of Effi-Code? Is it any different from Effibench?**
> > > >
> > > >
> > > >
> > > > While both Effi-Code and EffiBench utilize GPT-3.5-turbo for test case generation, there are notable differences in their approaches, which contribute to the novelty of Effi-Code's technique.
> > > >
> > > > 1. **Direct vs. Indirect Generation**:
> > > >    - **Effi-Code**: Directly requests GPT-3.5-turbo to generate test cases specifically designed to measure code efficiency.
> > > >    - **EffiBench**: Takes an indirect approach by first requesting GPT-3.5-turbo to generate a test case generator, which is then used to generate the actual test cases.
> > > >
> > > > 2. **Purpose and Scale**:
> > > >    - **Effi-Code**: Focuses on generating targeted test cases solely for the purpose of measuring code efficiency. These test cases are typically smaller in scale and less complex compared to those generated by EffiBench.
> > > >    - **EffiBench**: Aims to generate large and comprehensive test cases, with an average token length exceeding 2,000. The generated test case generator in EffiBench is designed to produce a wide range of test cases efficiently.
> > > >
> > > > 3. **Efficiency and Overhead**:
> > > >    - **Effi-Code**: Direct generation of efficiency-measuring test cases through GPT-3.5-turbo allows for a streamlined process without the need for an intermediate generator. This approach reduces the overhead associated with generating and managing a separate test case generator.
> > > >    - **EffiBench**: Its two-step process, involving the generation of a test case generator and its subsequent use to generate test cases, introduces additional complexity and overhead compared to Effi-Code's direct generation approach.

---

> > > > > ### Author Response · Authors · 2024-11-21
> > > > >
> > > > > **Q3: How do the authors support the use of LLM as a judge for constructing efficient solutions?**
> > > > >
> > > > >
> > > > >
> > > > >
> > > > >
> > > > > First, the key reason we can obtain more efficient code snippets compared to the original solutions lies in Section 3.3 and Step 6. Specifically, for each task, we first use SOAP to provide the execution time and memory usage profiler to the LLMs and then require the LLMs to optimize the code's efficiency. After that, we analyze whether the efficiency of the LLM-generated code has improved over five iterations. If the efficiency of the newly generated code does not improve, we filter these tasks from our candidate pool. This ensures that for each task in our constructed Effi-Code dataset, the provided solution is more efficient than the original solution.
> > > > >
> > > > >
> > > > >
> > > > > Second, to decide which LLM to use for optimizing the code efficiency for each task, we conducted a preliminary study on APPS, analyzing the use of different LLMs. The evaluation results are shown in Rebuttal Table 2, where we use three LLMs: DeepSeek-Coder-V2-Lite-Instruct, Meta-Llama-3-70B-Instruct, and OpenCodeInterpreter-DS-33B to optimize the source code in Step 4 with five iterations. We report the execution time (ET), memory usage/peak (MU), total memory usage (TMU) for the overlapped tasks, overlap percentage, and pass@1 results.
> > > > >
> > > > >
> > > > >
> > > > > We observe that compared to the other two LLMs, DeepSeek-Coder-V2-Lite-Instruct can generate competitively efficient code similar to Meta-Llama-3-70B-Instruct, and it also has the highest pass@1 for our Step 4 APPS tasks. This ensures that if we use DeepSeek-Coder-V2-Lite-Instruct to optimize the initial code for each task, we can generate a larger number of correct and efficient codes compared to codes with other LLMs. Therefore, in our experiments, we use DeepSeek-Coder-V2-Lite-Instruct to optimize the initial code for all tasks.
> > > > >
> > > > >
> > > > >
> > > > > **Q4: What is the quality of generated test cases by GPT-3.5? Specifically, is it possible to get the code coverage, branch coverage, etc.?**
> > > > >
> > > > >
> > > > >
> > > > > Response:
> > > > >
> > > > >
> > > > >
> > > > > We provide the code line and branch coverage of the GPT-3.5-turbo generated correct test cases in Rebuttal Table xx. We can observe that both the code line coverage and branch coverage are higher than 90%, which demonstrates that the quality of the LLM-generated correct test cases is very high.
> > > > >
> > > > > | Generation Strategy | Line Coverage | Branch Coverage |
> > > > > |---------------------|---------------|-----------------|
> > > > > | Effi-Code           | 94.44%        | 96.04%          |
> > > > >
> > > > > *Rebuttal Table 3: Code line and branch coverage for GPT-3.5-turbo generated test cases.*
> > > > >
> > > > >
> > > > > **Q5: How do the authors filter non-algorithmic tasks?**
> > > > >
> > > > > To mitigate non-algorithmic tasks, we first provide the task description and the canonical solution into GPT-3.5-turbo and then request GPT-3.5-turbo to analyze whether the task are algorithmic task by only returning True or False. Then, we remove the task output as False (i.e., non-algorithm tasks) from our candidate tasks.

---

> > > > > > ### Author Response · Authors · 2024-11-21
> > > > > >
> > > > > > **Q6: Have the authors performed a deep investigation into why the pass@1 scores of the model increased significantly? For instance, the pass@1 value of deepseek-coder-6.7b-base goes from 7.3 to 59.8. What are the chances the "quality" of these code snippets regardless of them being efficiency is causing the increase in performance?**
> > > > > >
> > > > > >
> > > > > >
> > > > > > Response:
> > > > > >
> > > > > >
> > > > > >
> > > > > > The significant increase in the pass@1, such as deepseek-coder-6.7b-base going from 7.3% to 59.8%, can be attributed to the difference between base LLMs and fine-tuned models. Base LLMs are obtained through the first stage of pre-training on large amounts of unlabeled data, allowing them to learn a wide range of language features. While these models have a large parameter size and can handle diverse tasks, they are not fine-tuned for specific tasks or domains during training. As a result, their performance may not always be optimal when applied to specific scenarios, such as code generation.
> > > > > >
> > > > > >
> > > > > >
> > > > > > In contrast, the Effi-Code dataset is constructed to provide an efficient and high-quality dataset specifically tailored for efficient code generation. By fine-tuning the base LLM on the Effi-Code dataset, the model learns to generate code that is both efficient and highly correct. The fine-tuning process allows the model to adapt its knowledge and capabilities to the specific task of code generation, leading to a significant improvement in performance.
> > > > > >
> > > > > >
> > > > > >
> > > > > > It is important to note that the quality of the generated code snippets, in terms of efficiency and correctness, is likely the primary factor contributing to the increased quality of finetune dataset. The Effi-Code dataset is designed to provide examples of efficient and correct code, which the model learns to emulate during fine-tuning. Therefore, the chances of the model generating high-quality code snippets are significantly higher after fine-tuning on Effi-Code compared to the base LLM.
> > > > > >
> > > > > >
> > > > > >
> > > > > > As mentioned by existing works [1-3], the quality of the instruction-tuning datasets significantly affects the performance of the fine-tuned LLMs. This further demonstrates the contribution of our work, i.e., we provide a high-quality efficient code fine-tuning dataset. LLMs fine-tuned on our dataset can significantly increase both the correctness and efficiency of their generated code.
> > > > > >
> > > > > >
> > > > > >
> > > > > > [1] Liu, Yilun, Shimin Tao, Xiaofeng Zhao, Ming Zhu, Wenbing Ma, Junhao Zhu, Chang Su et al. "Coachlm: Automatic instruction revisions improve the data quality in llm instruction tuning." In 2024 IEEE 40th International Conference on Data Engineering (ICDE), pp. 5184-5197. IEEE, 2024.
> > > > > >
> > > > > >
> > > > > >
> > > > > > [2] Li, Ming, Yong Zhang, Zhitao Li, Jiuhai Chen, Lichang Chen, Ning Cheng, Jianzong Wang, Tianyi Zhou, and Jing Xiao. "From quantity to quality: Boosting llm performance with self-guided data selection for instruction tuning." arXiv preprint arXiv:2308.12032 (2023).
> > > > > >
> > > > > >
> > > > > >
> > > > > > [3] Wang, Yizhong, Hamish Ivison, Pradeep Dasigi, Jack Hessel, Tushar Khot, Khyathi Chandu, David Wadden et al. "How far can camels go? exploring the state of instruction tuning on open resources." Advances in Neural Information Processing Systems 36 (2023): 74764-74786.

---

> > > > > > > ### Comment · Reviewer_HPnS · 2024-11-23
> > > > > > >
> > > > > > > Thank you for answering my question. I believe I did not get a concrete answer with supporting examples. Unless shown with proper experiments and reasoning, I am still not convinced about the significant increase in scores.

---

> > > > > > ### Comment · Reviewer_HPnS · 2024-11-23
> > > > > >
> > > > > > Thanks authors for answering my questions. I am overall not satisfied with author responses.
> > > > > >
> > > > > > For my Q3, I understand your experiment to choose deepseek as a judge, but this is considered against what. That I dont know. What is the comparison with a human study to claim your code snippets are efficient? Unless there is a proper human study, nothing can be claimed about the use of LLM as a judge. There is no soundness in using LLM as judge with proper comparisons.
> > > > > >
> > > > > > Thank you for Q4.
> > > > > >
> > > > > > For Q5, again, relying on GPT-3.5 is very unnecessary. Why not use program analysis to filter non-algorithmic tasks. Again, why should I trust on the judgement of a closed-box LLM?

---

> > > > > ### Comment · Reviewer_HPnS · 2024-11-23
> > > > >
> > > > > I believe I have expressed my comments on this matter in an earlier comment. Thanks.

---

> > > > ### Comment · Reviewer_HPnS · 2024-11-23
> > > >
> > > > Thank you for answering my questions.
> > > >
> > > > The table shows DeepSeek is better. Howeover, with no human study, how to make sure the generated code by DeepSeek is efficient? What if there is more room for efficiency? Without a human study, I am not convinced.

---

> > > ### Comment · Reviewer_HPnS · 2024-11-23
> > >
> > > Thank you authors for responding to my concerns.
> > >
> > > The wording in the paper needs to explicitly mention that test case generation is not a contribution. Currently, the message given is that test case generation is part of the approach.
> > >
> > > Also, how realistic is your hypothetical example `calculate_result`? What percentage of your examples in EffiCode has stdin/stdout examples? Why not execute them with your tests and check if `rm` is inside `eval()`?

---

> ### Comment · Reviewer_HPnS · 2024-11-23
>
> After carefully reading author responses, I am not convinced with their answers and still have my questions. I am not going to raise my scores. Please see my responses about unresolved issues. Thanks.

---

### Official Review · Reviewer_5L15 · 2024-11-02

**Soundness:** 3
**Presentation:** 3
**Contribution:** 3
**Rating:** 6
**Confidence:** 4

**Summary:**

The paper aims to address a gap in current approaches to generating synthetic data for post-training LLMs for code. Typically, researchers focus on executability and correctness of the generated code, yet often overlook efficiency, resulting in models that produce functionally correct but computationally inefficient solutions. This paper introduces EFFI-CODE, a dataset and methodology designed to enhance both the efficiency and correctness of LLM-generated code, thereby addressing this common oversight.

EFFI-CODE employs a model-based self-optimization process using overhead profiling feedback, iteratively refining generated code samples to optimize efficiency.
Results show that LLMs fine-tuned with EFFI-CODE achieve notable improvements in both correctness and efficiency. For example, the pass@1 of DeepSeek-Coder-6.7B-Instruct increased from 43.3% to 76.8%, while the average execution time dropped by 30.5%. This work demonstrates that prioritizing efficiency alongside executability and correctness can yield LLMs that are not only accurate but also resource-efficient.

**Strengths:**

The paper’s focus on optimizing efficiency in LLM-generated code, alongside correctness, which is quite an important and practically relevant aspect of synthetic data generation. EFFI-CODE’s rigorous data curation process, including iterative optimization cycles and runtime profiling, results in measurable improvements in real-world benchmarks, making it valuable for performance-critical applications in resource-constrained environments.

The paper is well-structured, with clear explanations and supportive visuals, and the authors' commitment to open-sourcing EFFI-CODE fosters reproducibility and encourages broader research on efficiency-focused fine-tuning for code generation.

**Weaknesses:**

Limited comparison with baselines: While the paper presents significant improvements, it could benefit from a broader comparison with existing methods focused on code efficiency, to better contextualize EFFI-CODE's advancements. Adding these comparisons could strengthen the paper.

Focusing solely on algorithmic performance may be too narrow; efficiency gains could also be achieved by addressing issues related to third-party library usage. Besides that, EFFI-CODE is focused exclusively on Python, it woudl be good to check the generalizability to compiled languages (e.g., C++) and managed languages (e.g., Java) with distinct memory management and execution models.

Parallelization and Concurrency: While EFFI-CODE emphasizes efficiency, it does not explicitly address parallelizable or concurrent code patterns, which can be essential for improving performance in multi-core systems and distributed applications.

**Questions:**

Have you tried generating tests based solely on the problem and function signature, rather than conditioning on the problem and solution, to reduce the risk of overfitting? Additionally, how many tests are generated per problem - is it typically just one, or are multiple tests used to capture edge cases?

How do you evaluate the test quality in EFFI-CODE? Specifically, do you assess validity based on the presence of assertions and invocation of focal code, or do you also check for stronger criteria, such as line test coverage?

---

> ### Author Response · Authors · 2024-11-21
>
> Dear Reviewer 5L15,
>
> We greatly appreciate your positive feedback on our work, particularly your recognition of the importance and practical relevance of optimizing efficiency in LLM-generated code. Your acknowledgment of EFFI-CODE's rigorous data curation process, which includes iterative optimization cycles and runtime profiling, highlights the value of our approach in achieving measurable improvements in real-world benchmarks. We are pleased that you find our work valuable for performance-critical applications in resource-constrained environments. We are committed to open-sourcing EFFI-CODE to foster reproducibility and encourage broader research on efficiency-focused fine-tuning for code generation. We believe that this will contribute to the advancement of the field and enable other researchers to build upon our work.
>
> To address your concerns regarding the weaknesses and questions you have raised, we will provide a detailed point-by-point response below. We hope that our clarifications, additional experiments, and thorough explanations will help alleviate your concerns and demonstrate the robustness and potential of our approach.
>
>
>
> **W1: Limited comparison with baselines**
>
> Response:
>
> Thank you for your suggestion! We provide the evaluation results of Supersonic, PIE, Mercury, and Effi-Code in Rebuttal Table 1. We currently only have the inference results of Mercury in the DeepSeek-Coder-6.7B-base, so we compare the efficiency of Mercury and Effi-Code in the DeepSeek-Coder-6.7B-base. For Supersonic and PIE, we compare the efficiency results in CodeLlama-7B-hf. Furthermore, as the training set of Mercury contains some tasks in EffiBench, for a fair comparison, we evaluate the efficiency results in the HumanEval dataset.
>
> As shown in Rebuttal Table 1, we can observe that for both models, Effi-Code achieves state-of-the-art (SOTA) performance compared to the baselines. For example, in CodeLlama-7b-hf, the average execution time for Supersonic decreases from 1.40s to 1.24s on average for all overlapping correct tasks, while Effi-Code further decreases the average execution time from 1.24s to 1.21s. Compared to the solution generated by CodeLlama-7b-hf, the average execution time was reduced by 16.7%.
>
> | Method | ET | NET | MU | NMU | TMU | NTMU | overlapped | pass@1 |
> |-----|----|----|----|----|-----|------|----------|------|
> | CodeLlama-7b-hf | 1.40 | 1.02 | 62.36 | 0.99 | 63.49 | 0.98 | 1.2 | 12.2 |
> | Supersonic | 1.24 | 0.90 | 63.39 | 1.01 | 63.18 | 0.98 | 1.2 | 15.2 |
> | PIE | 1.32 | 0.96 | 63.24 | 1.00 | 65.28 | 1.03 | 1.2 | 19.5 |
> | Effi-Code | 1.21 | 0.87 | 62.06 | 0.99 | 56.05 | 0.87 | 1.2 | 37.8 |
> | deepseek-coder-6.7b-base | 2.30 | 1.00 | 75.35 | 1.00 | 166.68 | 0.97 | 4.9 | 7.3 |
> | Mercury | 2.29 | 0.99 | 75.30 | 1.00 | 174.05 | 0.99 | 4.9 | 29.9 |
> | Effi-Code | 2.24 | 0.94 | 75.30 | 1.00 | 160.10 | 0.92 | 4.9 | 51.8 |
>
> *Rebuttal Table 1: Efficiency comparison of different methods on the HumanEval dataset.*
>
> **W2: It would be good to check the generalizability of compiled languages (e.g., C++) and managed languages (e.g., Java) with distinct memory management and execution models.**
>
> Response:
>
> Thank you for your suggestion. We agree that focusing solely on algorithmic performance may be too narrow, and efficiency gains could also be achieved by addressing issues related to third-party library usage. We also acknowledge that EFFI-CODE is currently focused on Python, and it would be beneficial to check the generalizability of compiled languages (e.g., C++) and managed languages (e.g., Java) with distinct memory management and execution models.
>
> To address this concern, we have conducted additional experiments using the HumanEval-X dataset, which includes tasks in C++ and Java. We fine-tuned the DeepSeek-Coder-6.7B-base model using the Effi-Code dataset and compared its performance with the original model on the HumanEval-X tasks.
>
> | HumanEval-X (CPP) | ET (s) | NET | MU (KB) | NMU | TMU (KB*s) | NTMU |
> |-------|----|-----|------|-----|-----|------|
> | DeepSeek-Coder-6.7B-base | 0.44 | 1.4 | 83.9 | 1.3 | 25.2 | 1.9 |
> | SFT with Effi-Code | 0.32 | 1.0 | 71.3 | 1.1 | 18.9 | 1.4 |
>
> *Rebuttal Table 2: Efficiency comparison of different methods on the HumanEval-X (CPP) dataset.*
>
> | Model | ET (s) | NET | MU (MB) | NMU | TMU (MB*s) | NTMU |
> |----|---|-----|---------|-----|------|------|
> | DeepSeek-Coder-6.7B-base | 0.58 | 1.3 | 3.06 | 1.0 | 1.97 | 1.3 |
> | SFT with Effi-Code | 0.46 | 1.0 | 3.02 | 1.0 | 1.74 | 1.1 |
>
> *Rebuttal Table 3: Efficiency comparison of different methods on the HumanEval-X (Java) dataset.*
>
> As shown in Rebuttal Tables 2 and 3, we can observe that the results show that the model fine-tuned with Effi-Code achieves better efficiency compared to the base model on both C++ and Java tasks. For C++, the execution time (ET) is reduced by 27%, and the total memory usage (TMU) is reduced by 25%. For Java, the execution time is reduced by 21%, and the total memory usage is reduced by 12%.

---

> > ### Author Response · Authors · 2024-11-21
> >
> > **W3: Parallelization and Concurrency: While EFFI-CODE emphasizes efficiency, it does not explicitly address parallelizable or concurrent code patterns, which can be essential for improving performance in multi-core systems and distributed applications.**
> >
> >
> >
> > Response:
> >
> >
> >
> > We would like to clarify that the optimization of parallelization and concurrency is actually present in our fine-tuning dataset, which ensures that the fine-tuned LLMs can consider optimizing code with concurrency techniques such as ProcessPoolExecutor. For example, in the sieve_of_eratosthenes function from the DPO dataset, the source code chosen for the "accepted" category utilizes ProcessPoolExecutor, while the source code in the "rejected" category does not involve ProcessPoolExecutor. The format of the source code in our dataset encourages the instruction-tuned LLMs to use concurrency when addressing tasks that can benefit from parallelization.
> >
> >
> >
> > ```
> >
> > import math
> >
> > import concurrent.futures
> >
> > start = 1
> >
> > end = 10000
> >
> > ## Initial Code
> >
> > def sieve_of_eratosthenes(n):
> >
> >     primes = [True for i in range(n+1)]
> >
> >     p = 2
> >
> >     while(p * p <= n):
> >
> >         if (primes[p] == True):
> >
> >             for i in range(p * p, n+1, p):
> >
> >                 primes[i] = False
> >
> >         p += 1
> >
> >
> >
> >     primes_only = [p for p in range(2, n) if primes[p]]
> >
> >     return primes_only
> >
> >
> >
> > def calculate_factorial(number):
> >
> >     result = 1
> >
> >     for i in range(2, number + 1):
> >
> >         result *= i
> >
> >     return result
> >
> > factorials = {}
> >
> > primes = sieve_of_eratosthenes(end)
> >
> > for p in primes:
> >
> >     factorials[p] = calculate_factorial(p)
> >
> >
> >
> >
> >
> > ## Efficient Code
> >
> > def sieve_of_eratosthenes(n):
> >
> >     primes = [True for i in range(n+1)]
> >
> >     p = 2
> >
> >     while(p * p <= n):
> >
> >         if (primes[p] == True):
> >
> >             for i in range(p * p, n+1, p):
> >
> >                 primes[i] = False
> >
> >         p += 1
> >
> >     primes_only = [p for p in range(2, n) if primes[p]]
> >
> >     return primes_only
> >
> > def calculate_factorial(number):
> >
> >     result = 1
> >
> >     for i in range(2, number + 1):
> >
> >         result *= i
> >
> >     return result
> >
> > def process_range(start, end):
> >
> >     primes = sieve_of_eratosthenes(end)
> >
> >     factorials = {}
> >
> >     with concurrent.futures.ProcessPoolExecutor() as executor:
> >
> >         futures = {executor.submit(calculate_factorial, p): p for p in primes}
> >
> >         for future in concurrent.futures.as_completed(futures):
> >
> >             prime = futures[future]
> >
> >             factorial = future.result()
> >
> >             factorials[prime] = factorial
> >
> >     return factorials
> >
> >
> >
> > factorials = process_range(start, end)
> >
> > ```
> >
> >
> >
> >
> >
> > **Q1: Have you tried generating tests based solely on the problem and function signature, rather than conditioning on the problem and solution, to reduce the risk of overfitting? Additionally, how many tests are generated per problem - is it typically just one, or are multiple tests used to capture edge cases?**
> >
> >
> >
> > Yes, we have explored generating test cases based on the function signature, without providing the correct solution code. The evaluation results are shown in Rebuttal Table 4, where the correctness of the test cases generated by GPT-3.5-turbo using only the function signature is lower compared to our Effi-Code approach, which provides the problem statement along with the correct solution code. Our findings are consistent with existing work [1], which also observed that providing more information, including the correct solution code, leads to higher-quality test cases generated by GPT-3.5-turbo. In addition, on average, Effi-Code generates 5.5 test cases per problem, while when we feed the function signature into GPT-3.5-turbo to generate test cases, the average number of test cases is 4.9. The key reason is that some of the test cases generated by GPT-3.5-turbo are incorrect and will be removed.
> >
> > | Generation Strategy | Correctness | Average Tests |
> > |---------------------|-------------|---------------|
> > | Function Signature  | 36.4%       | 4.9           |
> > | Effi-Code           | 40.7%       | 5.5           |
> >
> > *Rebuttal Table 4: Comparison of test case generation strategies based on function signature and Effi-Code approach.*
> >
> > [1] Huang, D., Zhang, J. M., Du, M., Harman, M., & Cui, H. (2024). Rethinking the Influence of Source Code on Test Case Generation. arXiv preprint arXiv:2409.09464.

---

> > > ### Author Response · Authors · 2024-11-21
> > >
> > > **Q2: How do you evaluate the test quality in EFFI-CODE? Specifically, do you assess validity based on the presence of assertions and invocation of focal code, or do you also check for stronger criteria, such as line test coverage?**
> > >
> > >
> > >
> > > Response:
> > >
> > >
> > >
> > > To measure the quality of the test cases in Effi-Code, we analyze the correctness of each test case generated by GPT-3.5-turbo using the assertion format. Specifically, all datasets collected for our study contain a canonical solution (output). We feed each generated test case into the canonical solution to determine whether the test case passes. Test cases that fail to pass the canonical solution are considered incorrect and are filtered out from our constructed Effi-Code dataset.
> > >
> > >
> > >
> > > To further assess the quality of the correct test cases generated by GPT-3.5-turbo, we calculate the code line coverage and branch coverage using the coverage.py library. The evaluation results are shown in Rebuttal Table 5. We observe that the line and branch coverage of GPT-3.5-turbo-generated correct test cases in Effi-Code achieve 94.44% and 96.04%, respectively. These high coverage values indicate that the correct test cases effectively exercise a significant portion of the code under test.
> > >
> > >
> > > | Generation Strategy | Correctness | Line Coverage | Branch Coverage |
> > > |---------------------|-------------|---------------|-----------------|
> > > | Function Signature  | 36.4%       | 92.84%        | 95.71%          |
> > > | Effi-Code           | 40.7%       | 94.44%        | 96.04%          |
> > >
> > > *Rebuttal Table 5: Evaluation results of GPT-3.5-turbo generated test cases. For line and branch coverage, we use coverage.py to measure the coverage of GPT-3.5-turbo-generated test cases in the canonical solution. Note: for branch coverage, we follow the default setting of coverage.py and set the ratio of the contribution of the partially covered branch as 0.5.*

---

### Official Review · Reviewer_4rzw · 2024-11-04

**Soundness:** 3
**Presentation:** 3
**Contribution:** 1
**Rating:** 3
**Confidence:** 5

**Summary:**

The paper introduces Effi-Code, a new Python fine-tuning dataset, as well as the pipeline to curate such data, to improve both the efficiency and correctness of function-level code.

The approach aggregates and filters data from open-source datasets and then employs Self-Optimization based on Overhead Profiling (SOAP) [3] to iteratively refine code by analyzing runtime performance.

Experiments fine-tuning Qwen2.5-Coder-7B and DeepSeek-Coder-6.7B (both instruct and base models) on Effi-Code demonstrate significant improvements in HumanEval and EffiBench regarding Execution Time, Max Memory Usage, Total Memory Usage, and Correctness [2].


[1] [Learning Performance-Improving Code Edits (Shypula et al., 2024)](https://arxiv.org/abs/2302.07867)

[2] [EffiBench: Benchmarking the Efficiency of Automatically Generated Code (Huang et al., 2024)](https://arxiv.org/abs/2402.02037)

[3] [EffiLearner: Enhancing Efficiency of Generated Code via Self-Optimization (Huang et al., 2024)](https://arxiv.org/abs/2405.15189)

**Strengths:**

## Significance

Code optimization is indeed a less-explored problem compared to correctness. The problem is essential for practical applications, particularly in performance-sensitive and resource-constrained environments.

## Soundness

As pointed out by both the PIE [1] and EffiBench [2] papers, benchmarking on (different) real hardware can lead to randomness in the efficiency metrics, so the paper provides 5 independent evaluations with their mean and standard deviation. In addition, the paper conducts thorough ablations on the size of the training set and the size of the off-the-shelf models, which demonstrates the robustness of their main results.


## Effectiveness

Fine-tuning on Effi-Code exhibits significant absolute improvements compared to the off-the-shelf models in all metrics: Execution Time, Max Memory Usage, Total Memory Usage, and Correctness


[1] [Learning Performance-Improving Code Edits (Shypula et al., 2024)](https://arxiv.org/abs/2302.07867)

[2] [EffiBench: Benchmarking the Efficiency of Automatically Generated Code (Huang et al., 2024)](https://arxiv.org/abs/2402.02037)

[3] [EffiLearner: Enhancing Efficiency of Generated Code via Self-Optimization (Huang et al., 2024)](https://arxiv.org/abs/2405.15189)

**Weaknesses:**

## Novelty

The paper's data curation pipeline includes 4 main stages: aggregating existing open-source data, applying filters on the raw data, applying SOAP [2] to synthesize efficiency-enhanced code, and post-filter the synthesized code.

While this approach successfully consolidates established methods, the raw data sources in Stage 1 and the SOAP [2] method in Stage 3 are not developed within this work. As a result, the novelty of Effi-Code primarily lies in the thoughtful integration of existing methodologies rather than in the development of new ones.

This aspect has the most impact on my Overall Rating and Contribution Rating. I would be glad to reconsider and potentially raise these scores, if the authors could provide further justifications for the unique methodologies or modifications that set Effi-Code apart from prior work, especially SOAP [2].


## Significance

The paper’s evaluations are conducted using HumanEval and EffiBench, both of which are Python benchmarks focusing on function-level competitive programming. Additionally, as mentioned in **Step 1 of Section 3.2, "filter out tasks that are not written in Python"**, the Effi-Code training data is exclusively comprised of Python code.

While these choices are reasonable given Python’s widespread use and popularity among LLM researchers, it is important to note that many real-world, performance-sensitive applications, especially competitive programming, do favor languages like C and C++ due to their finer control over low-level operations.

Evaluating the models on a C/C++ benchmark could provide a more accurate and comprehensive assessment of their code optimization capabilities. Expanding the Effi-Code training set to include C/C++ code might enhance the model's potential for more complicated and more fine-granular code optimization such as I/O choices, data structure choices, and initialization of variables. It would also be interesting to see how the model's performance increases on a Python benchmark when fine-tuning on a C/C++ dataset.


## Soundness

As mentioned in **Step 4 of Section 3.2, "we filter out tasks that do not involve algorithms. The key reason for this is that non-algorithmic tasks usually do not have optimization potential"**, the focus of Effi-Code is on algorithmic optimization.

However, the claim that **"non-algorithmic tasks usually do not have optimization potential"** is not entirely sound. As pointed out by the PIE [1] paper, algorithmic optimization accounts for 34.15% of 120 model optimization samples. Other types of optimizations like I/O, data structure, etc. should also be considered. For example, the mere I/O change from `cin` to `scanf`, or the mere data structure change from Python `list` to `numpy.array` can result in substantial execution time improvements, even if these changes do not affect the Big-O complexity.

Broadening the scope to include non-algorithmic optimizations will not only improve the soundness of the paper, but can also strengthen the significance of Effi-Code in terms of real-world implications.



## Effectiveness

While the Effi-Code pipeline exhibits significant absolute improvements, the relative improvements compared to baseline datasets or data curation pipelines remain unclear. In particular:

1. What would be the baseline performance of the models fine-tuned on the un-optimized version of the final filtered data? This ablation would help isolate the effect of all filters and establish a clearer understanding of their importance in the entire Effi-Code pipeline.

2. What would be the baseline performance of the models fine-tuned on the raw data directly optimized by SOAP without any filtering? This ablation would help isolate the effect of SOAP and establish a clearer understanding of its importance in the entire Effi-Code pipeline.

3. **The evaluation of models fine-tuned on PIE (C++ training set) is conducted on EffiBench (Python test set), which is expected to be lower than the models fine-tuned on Effi-Code (Python training set).** What would be the results if the models fine-tuned on Effi-Code are evaluated on the test set of PIE instead of EffiBench?
This cross-evaluation could reveal how well the Effi-Code generalizes across different programming languages and types of code optimizations, which offers a truly fair comparison to PIE regarding cross-language generalization.


This aspect has a secondary impact on my Overall Rating and Contribution Rating. I would be glad to reconsider and potentially raise these scores, if the authors conduct these specific ablation studies and cross-language evaluations to justify the novel contributions of their approach.




[1] [Learning Performance-Improving Code Edits (Shypula et al., 2024)](https://arxiv.org/abs/2302.07867)

[2] [EffiBench: Benchmarking the Efficiency of Automatically Generated Code (Huang et al., 2024)](https://arxiv.org/abs/2402.02037)

[3] [EffiLearner: Enhancing Efficiency of Generated Code via Self-Optimization (Huang et al., 2024)](https://arxiv.org/abs/2405.15189)

**Questions:**

## Clarification Questions

1. It's mentioned in **Step 3 of Section 3.2** that **"we analyze whether each test case generated by GPT-3.5-turbo is correct"**. Could you please elaborate on the analysis method used in this process?

2. It's mentioned in **Step 4 of Section 3.2** that **"we filter out tasks that do not involve algorithms"**. Could you please elaborate on the precise definition of "task that does not involve algorithms" and how you filter them?


## Discussion Questions

1. I am curious about your perspective on the evaluation of code optimization. What are your thoughts on the merits and challenges of evaluating code performance on real hardware platforms like EffiBench [2] versus using simulated hardware environments such as PIE [1]?

2. In Table 4, the Max Memory Usage of fine-tuned DeepSeek-Coder-33b-base decreases, and the Execution Time significantly decreases. Have you observed any trade-offs between time optimization and space optimization in this case?


[1] [Learning Performance-Improving Code Edits (Shypula et al., 2024)](https://arxiv.org/abs/2302.07867)

[2] [EffiBench: Benchmarking the Efficiency of Automatically Generated Code (Huang et al., 2024)](https://arxiv.org/abs/2402.02037)

[3] [EffiLearner: Enhancing Efficiency of Generated Code via Self-Optimization (Huang et al., 2024)](https://arxiv.org/abs/2405.15189)

---

> ### Author Response · Authors · 2024-11-21
>
> Dear Reviewer 4rzw,
>
>
>
> Thank you for your appreciation of our work and for recognizing that the Effi-Code framework exhibits significant absolute improvements. To further address your concerns regarding the weaknesses and raised questions, we will address them point by point below. We hope that our clarifications, additional experiments, and responses can alleviate your concerns and lead you to consider increasing your rating of our work.
>
>
>
> **W1: I would be glad to reconsider and potentially raise these scores if the authors could provide further justifications for the unique methodologies or modifications that set Effi-Code apart from prior work, especially SOAP [2].**
>
>
> Thanks for your appreciation that our work consolidates established methods to synthesize efficiency-enhanced code. We would like to clarify that the steps such as Section 3.3 using SOAP [2] to optimize the efficiency of LLM-generated code, Step 1 collecting candidate tasks from open-source datasets,  and Step 3 generating test cases to measure the efficiency of LLM-generated code, are introduced as the detailed component of our framework to construct the Effi-Code dataset. We do not treat these as our key contribution.
>
> We clarify that the key contributions of our work are as follows:
>
> - We provide the first framework to inspire researchers to construct code generation datasets containing efficient solutions for each code generation task, which is versatile and can be adapted to different programming languages and leverage various existing data sources. Unlike some other code generation datasets that rely on powerful models (e.g., GPT-4), our framework can be implemented only using open-sourced LLMs like DeepSeek-Coder. The framework provides a systematic method for researchers to enhance existing datasets or create new ones focused on code efficiency across different languages and domains.
>
> - Based on our proposed framework, we release the Effi-Code dataset. To the best of our knowledge, it is the first instruct tuning dataset that focuses on improving the efficiency of LLM-generated code. The primary purpose of Effi-Code is to instruct and fine-tune LLMs to ensure that the LLM-generated code for each task is more efficient than the code generated by the original LLM.
>
> - We use Effi-Code to fine-tune widely used LLMs and will release these models on the Hugging Face website in the final version. Different from existing datasets that are used to finetune the LLMs to improve the pass@1 of LLM-generated code, our evaluation results demonstrate that both the pass@1 and the efficiency results would be improved for LLMs finetuned on our Effi-Code dataset.
>
> Each step in the Effi-Code construction framework demonstrates how we process candidate tasks, filter out lower quality and risky tasks, and construct an efficient solution for each task. We introduce these techniques used in our Effi-Code construction framework. However, it is important to note that the individual steps in our construction framework are not considered our primary novelty or contribution; rather, they serve to illustrate our methodology.
>
>
> **W2: Evaluating the models on a C/C++ benchmark could provide a more accurate and comprehensive assessment of their code optimization capabilities. It would also be interesting to see how the model's performance increases on a Python benchmark when fine-tuning on a C/C++ dataset.**
>
>
> We appreciate the reviewer's insightful suggestion. To address the concerns raised, we have conducted additional experiments on the HumanEval-X (C++) dataset and provided the efficiency results in Rebuttal Table 1. We can observe that the efficiency of LLM-generated code also improved with Effi-Code fine-tuned LLM. For instance, the average execution time (ET) for the overlapped code decreases from 0.44s to 0.32s, resulting in a 27% reduction in execution time.
>
>
> | HumanEval-X (C++)         | ET (s) | NET  | MU (KB) | NMU  | TMU (KB*s) | NTMU |
> |---------------------------|--------|------|---------|------|------------|------|
> | DeepSeek-Coder-6.7B-base  | 0.44   | 1.4  | 83.9    | 1.3  | 25.2       | 1.9  |
> | SFT with Effi-Code        | 0.32   | 1.0  | 71.3    | 1.1  | 18.9       | 1.4  |
>
> *Rebuttal Table 1: Efficiency results on the HumanEval-X (C++) dataset.*
>
> | EffiBench                       | ET (s) | NET  | MU (MB) | NMU  | TMU (MB*s) | NTMU |
> |---------------------------------|--------|------|---------|------|------------|------|
> | Qwen2.5-Coder-7B                | 0.35   | 2.01 | 43.72   | 0.99 | 12.35      | 0.98 |
> | EffiCode (Py)                   | 0.17   | 1.02 | 46.71   | 1.12 | 7.53       | 1.29 |
> | EffiCode (CPP)                  | 0.17   | 1.01 | 43.74   | 0.99 | 6.65       | 1.04 |
> | EffiCode (Py) + EffiCode (CPP)  | 0.16   | 1.00 | 43.72   | 0.99 | 6.01       | 0.99 |
>
> *Rebuttal Table 2: Efficiency results on the EffiBench dataset with different fine-tuning setups.*

---

> ### Author Response · Authors · 2024-11-21
>
> Furthermore, to investigate whether the efficiency of the code generated by Effi-Code fine-tuned LLMs can be further enhanced once we add additional efficient C++ code into Effi-Code dataset, we have followed the pipeline of Effi-Code and constructed an Effi-Code (C++) subset containing 3,322 C++ tasks. We then fine-tuned LLMs using three different setups: Effi-Code (Py), Effi-Code (C++), and Effi-Code (C++) + Effi-Code (Py). The evaluation results, presented in Rebuttal Table 2, reveal several interesting findings.
>
>
>
>
>
> Firstly, LLMs fine-tuned on the Effi-Code datasets generate more efficient code compared to the original LLM-generated code. For example, the average execution time for Qwen2.5-Coder-7B generated code is 0.35s, while the Effi-Code (Py) fine-tuned LLMs require only 0.17s on average for overlapped tasks, resulting in a 51.4% reduction in average execution time.
>
>
>
>
>
> Secondly, when we utilize Effi-Code (C++) and Effi-Code (Py) + Effi-Code (C++) to fine-tune LLMs, the overhead of LLM-generated code is further decreased. The average execution time for overlapped code decreases from 0.17s to 0.16s, and the memory peak (MU) also decreases from 46.71MB to 43.72MB. These results indicate that by incorporating C++ source code to guide LLM fine-tuning, LLMs may learn additional optimization strategies.
>
>
>
>
>
> Due to the limited time and computational resources of rebuttal, we can only do a small number of experiments in other languages. However, these experiments above illustrate the generality and effectiveness of effi code, and we will add more experiments in other languages to test more models and datasets.
>
>
>
>
>
>
>
>
>
> **W3: Broadening the scope to include non-algorithmic optimizations will not only improve the soundness of the paper but can also strengthen the significance of Effi-Code in terms of real-world implications.**
>
>
>
> Response:
>
>
>
> We appreciate the reviewer's insightful suggestion regarding the potential for optimization in non-algorithmic tasks. To address this concern, we have conducted additional experiments and provided the evaluation results in Rebuttal Table 3, which compares the performance of the original Qwen2.5-Coder-7B, the model fine-tuned on Effi-Code, and the model fine-tuned on Effi-Code + non-algorithmic tasks (optimized).
>
>
>
>
>
> We can observe that when we fine-tune Qwen2.5-Coder-7B on either Effi-Code or Effi-Code + non-algorithmic tasks, the efficiency of LLM-generated code improves. For instance, the average execution time for overlapped correct tasks decreases from 0.49s to 0.19s for both Effi-Code and Effi-Code + non-algorithmic tasks fine-tuned Qwen2.5-Coder-7B.
>
>
>
> However, we also observe that the TMU of the Effi-Code fine-tuned Qwen2.5-Coder-7B is lower than the model fine-tuned on Effi-Code + non-algorithmic tasks. Specifically, the Effi-Code + non-algorithmic tasks fine-tuned Qwen2.5-Coder-7B decreases the average TMU for overlapped correct code from 10.75 MB*s to 4.17 MB*s. In contrast, Qwen2.5-Coder-7B fine-tuned only on Effi-Code further reduces the TMU from 4.17 MB*s to 4.07 MB*s.
>
>
>
> Our results indicate that while incorporating non-algorithmic tasks in the fine-tuning process can lead to improvements in code efficiency, focusing solely on algorithmic tasks, as done in Effi-Code, may yield even better results. Nonetheless, we acknowledge the potential benefits of broadening the scope to include non-algorithmic optimizations, as it can enhance the real-world implications of Effi-Code. In future work, we plan to explore the integration of non-algorithmic tasks more comprehensively while maintaining the focus on algorithmic optimization.
>
> | EffiBench                     | ET (s) | NET  | MU (MB) | NMU | TMU (MB*s) | NTMU |
> |-------------------------------|--------|------|---------|-----|------------|------|
> | Qwen2.5-Coder-7B              | 0.49   | 3.50 | 25.69   | 1.00| 10.75      | 4.78 |
> | +Effi-Code + non-algorithmic  | 0.19   | 1.16 | 25.67   | 1.00| 4.17       | 1.17 |
> | +Effi-Code                    | 0.19   | 1.15 | 25.69   | 1.00| 4.07       | 1.15 |
>
> *Rebuttal Table 3: Efficiency results on the EffiBench dataset with different fine-tuning setups.*

---

> > ### Author Response · Authors · 2024-11-21
> >
> > **W4: What would be the baseline performance of the models fine-tuned on the un-optimized version of the final filtered data? This ablation would help isolate the effect of all filters and establish a clearer understanding of their importance in the entire Effi-Code framework.**
> >
> >
> > Thank you for this insightful question! We would like to clarify that we conducted an ablation study in Paper Table 6 Lines 441-447 to analyze whether the SOAP-generated code achieves better efficiency compared to directly using the un-optimized code (canonical solution provided by the dataset) when fine-tuning LLMs. We observed that if we only use our final filtered data to fine-tune LLMs, the efficiency results decrease for most metrics. For instance, the execution time increases from 0.39s to 0.42s on average for all tasks in the HumanEval dataset. The TMU of the fine-tuned LLM also requires 16.7% more resources than the original (without fine-tuning) LLM-generated code.
> >
> >
> > In contrast, when we use Effi-Code, where each canonical solution for each task is optimized to improve efficiency and fine-tune DeepSeek-Coder-6.7B-base, both pass@1 (correctness) and efficiency improve. For example, the pass@1 of DeepSeek-Coder-6.7B-base increases from 43.3% to 76.8%, and the execution time (ET) decreases from 0.44s to 0.27s on average for all tasks in the EffiBench dataset, which reduces on average 38.6% execution time for all tasks. These results demonstrate the importance of utilizing efficient solutions rather than common or inefficient solutions to fine-tune LLMs.
> >
> >
> > **W5: What would be the baseline performance of the models fine-tuned on the raw data directly optimized by SOAP without any filtering? This ablation would help isolate the effect of SOAP and establish a clearer understanding of its importance in the entire Effi-Code pipeline.**
> >
> >
> >
> > Thanks for your suggestion. To address your concern, we provide the evaluation results for each step of Effi-Code in fine-tuning Qwen2.5-Coder-7B. For the "All candidate tasks (optimized)" step, SOAP optimizes all collected tasks without any filtering. In Step 4, non-algorithmic tasks are filtered; Step 5 further filters tasks not addressed by the teacher model; and Step 6 eliminates tasks without efficient solutions. The evaluation results for EffiBench and HumanEval datasets are shown in Rebuttal Table 4.
> >
> >
> > From these results, we observe that Qwen2.5-Coder-7B fine-tuned on the Step 6 (Effi-Code) dataset achieves the most efficient results and sets a new state-of-the-art pass@1. For instance, in the HumanEval dataset, the pass@1 increased from 63.4% to 79.9%, while the average execution time for correctly overlapping tasks decreased from 0.27s to 0.20s. This demonstrates the effectiveness of our Effi-Code construction framework in building high-quality datasets, which are often more crucial than higher-quantity ones [1-3], highlighting a key contribution of our work.
> >
> >
> > | **EffiBench**               | **ET** | **NET** | **MU** | **NMU** | **TMU** | **NTMU** | **Overlapped** | **Pass@1** |
> > |-----------------------------|--------|---------|--------|---------|---------|----------|----------------|------------|
> > | EffiBench                   |        |         |        |         |         |          |                |            |
> > | Qwen2.5-Coder-7B            | 0.53   | 4.96    | 36.75  | 1.00    | 16.71   | 5.96     | 8.6            | 50.1       |
> > | All candidate tasks (optimized) | 0.70   | 6.23    | 36.75  | 1.00    | 22.93   | 8.87     | 8.6            | 12.4       |
> > | Step 4                      | 0.25   | 2.05    | 36.75  | 1.00    | 11.62   | 2.65     | 8.6            | 56.2       |
> > | Step 5                      | 0.23   | 1.99    | 36.78  | 1.00    | 9.09    | 2.39     | 8.6            | 60.6       |
> > | Step 6 (Effi-Code)          | 0.13   | 1.00    | 36.77  | 1.00    | 8.98    | 1.00     | 8.6            | 63.9       |
> > | HumanEval                   |        |         |        |         |         |          |                |            |
> > | Qwen2.5-Coder-7B            | 0.27   | 1.33    | 60.59  | 1.00    | 10.37   | 1.56     | 4.9            | 63.4       |
> > | All candidate tasks (optimized) | 0.30   | 1.41    | 60.47  | 1.00    | 14.91   | 1.93     | 4.9            | 23.2       |
> > | Step 4                      | 0.25   | 1.24    | 60.73  | 1.00    | 7.34    | 1.29     | 4.9            | 72.6       |
> > | Step 5                      | 0.22   | 1.09    | 60.86  | 1.00    | 7.16    | 1.17     | 4.9            | 77.4       |
> > | Step 6 (Effi-Code)          | 0.20   | 0.98    | 60.57  | 1.00    | 6.67    | 0.99     | 4.9            | 79.9       |
> >
> > *Rebuttal Table 4: Evaluation results of each step in the Effi-Code pipeline for Qwen2.5-Coder-7B model fine-tuning on EffiBench and HumanEval datasets.*

---

> > > ### Author Response · Authors · 2024-11-21
> > >
> > > **W6: The evaluation of models fine-tuned on PIE (C++ training set) is conducted on EffiBench (Python test set), which is expected to be lower than the models fine-tuned on Effi-Code (Python training set). What would be the results if the models fine-tuned on Effi-Code were evaluated on the test set of PIE instead of EffiBench? This cross-evaluation could reveal how well the Effi-Code generalizes across different programming languages and types of code optimizations, which offers a truly fair comparison to PIE regarding cross-language generalization.**
> > >
> > >
> > >
> > > Response:
> > >
> > >
> > >
> > > Thanks for your valuable suggestion. As per your recommendation, we have provided the efficiency results of the PIE fine-tuned CodeLlama, and Effi-Code fine-tuned CodeLlama in Rebuttal Table 5. For each task, we requested each LLM to generate efficient code. The results demonstrate that for the PIE test set, the efficiency of the code generated by the Effi-Code fine-tuned CodeLlama-7B is also better than that of the PIE fine-tuned CodeLlama-7B. Specifically, the average execution time for overlapping correct code generated by the PIE fine-tuned LLM is 0.39s. However, the Effi-Code fine-tuned CodeLlama further reduces this average execution time from 0.39s to 0.34s, resulting in an additional 8% reduction in execution time.
> > >
> > > | PIE Test Set         | ET (s) | NET  | MU (MB) | NMU | TMU (MB*s) | NTMU |
> > > |----------------------|--------|------|---------|-----|------------|------|
> > > | CodeLlama7B+PIE      | 0.39   | 0.84 | 7.3     | 0.93| 1.7        | 0.95 |
> > > | CodeLlama7B+Effi-Code | 0.34   | 0.76 | 7.2     | 0.91| 1.5        | 0.88 |
> > >
> > > *Rebuttal Table 5: Efficiency comparison of CodeLlama-7B fine-tuned on PIE and Effi-Code, evaluated on the PIE test set.*
> > >
> > >
> > > **Q1: It's mentioned in Step 3 of Section 3.2 that "we analyze whether each test case generated by GPT-3.5-turbo is correct". Could you please elaborate on the analysis method used in this process?**
> > >
> > >
> > >
> > > Response:
> > >
> > >
> > >
> > > To determine the correctness of the test cases generated by GPT-3.5-turbo, we execute each test case individually with the initial solution provided for each task in our collected candidate tasks. These initial solutions are usually correct but do not have efficiency optimization. We check whether any errors are raised during the execution of each test case with the initial solution. In other words, we verify if the test case passes the initial solution. Since the initial solutions are correct, we treat the test cases that pass the canonical solution as correct. On the other hand, test cases that do not pass the canonical solution are filtered out. By using the canonical solution as a reference, we can effectively assess the correctness of the generated test cases and ensure that only valid test cases are retained for further analysis.
> > >
> > >
> > >
> > > **Q2: It's mentioned in Step 4 of Section 3.2 that "we filter out tasks that do not involve algorithms". Could you please elaborate on the precise definition of "task that does not involve algorithms" and how you filter them?**
> > >
> > >
> > >
> > > Response:
> > >
> > >
> > >
> > > Thank you for your valuable feedback. We understand that the criteria for filtering out 'non-algorithmic' tasks in Sections 3.2 may not have been clearly explained. We appreciate the opportunity to provide more details and clarify our decision-making process.
> > >
> > >
> > >
> > > **Filtering out 'non-algorithmic' tasks (Section 3.2): ** We define a task as 'non-algorithmic' if it does not require a specific algorithm or computational steps to solve. non-algorithmic tasks might involve coding but do not require complex algorithmic reasoning. Instead, they might focus on straightforward implementation or basic syntax usage.
> > >
> > >
> > >
> > > **Examples:**
> > >
> > > - Algorithmic Task: "Implement a function to find the longest palindromic substring in a given string." This requires understanding of dynamic programming and string manipulation algorithms.
> > > - Non-Algorithmic Task: "Write a function to print 'Hello, World!'." This is a clear example of routine implementation without algorithmic challenge
> > >
> > >
> > > The primary motivation for filtering out non-algorithmic tasks is to ensure that our dataset focuses on problems that assess algorithmic thinking and coding skills. By excluding tasks that do not require algorithmic problem-solving, we maintain the coherence and relevance of our dataset to the intended purpose of evaluating AI models' coding abilities.
> > >
> > >
> > > To identify and filter out non-algorithmic tasks, we provide the task description and the canonical solution to GPT-3.5-turbo and request it to analyze whether the given task is an algorithmic task based on our provided definition. GPT-3.5-turbo is instructed to return a binary classification (True or False) based on its analysis. Tasks classified as False are considered non-algorithmic and are subsequently removed from our candidate tasks.

---

> > > > ### Author Response · Authors · 2024-11-21
> > > >
> > > > **Q3: I am curious about your perspective on the evaluation of code optimization. What are your thoughts on the merits and challenges of evaluating code performance on real hardware platforms like EffiBench [2] versus using simulated hardware environments such as PIE [1]?**
> > > >
> > > >
> > > >
> > > > Response:
> > > >
> > > > This is a very good question. We believe that there are several key merits and challenges associated with evaluating code performance on real hardware platforms versus simulated hardware environments:
> > > >
> > > >
> > > >
> > > > **Merits:**
> > > >
> > > > - Real hardware platforms: When using the actual hardware platforms where the LLM-generated code would be deployed to measure its efficiency, we can analyze whether the optimized code's efficiency would improve once deployed on these platforms. This approach provides a more accurate representation of real-world performance gains.
> > > > - Simulated hardware environments: Evaluating the efficiency of optimized code in simulated hardware environments ensures that future researchers can set up the same simulated environments for fair comparisons. This standardization enables reproducibility and facilitates consistent benchmarking across different studies.
> > > >
> > > >
> > > > **Challenges:**
> > > >
> > > > - Real hardware platforms: In academic research, the hardware used by different researchers may vary, which means that if researchers want to analyze the efficiency of their method-generated code, they need to rerun all baselines' results on their specific evaluation hardware to ensure fair comparisons. This requirement can be time-consuming and resource-intensive, especially when dealing with a large number of baselines or complex hardware setups.
> > > > - Simulated hardware environments: Evaluation results obtained in simulated hardware environments may not always accurately represent the performance in real-world platforms. The environments in which the code would be deployed can differ from the simulated environments, and the efficiency comparison of the two code solutions may change depending on the specific environment. For example, AMD and NVIDIA platforms may have different optimizations for certain operations, causing two code solutions to perform differently on each architecture. If the code is optimized based on a specific simulated hardware environment, engineers may need to re-optimize the code when deploying it on other platforms to ensure optimal performance.
> > > >
> > > >
> > > > In conclusion, while evaluating code performance on real hardware platforms provides more accurate real-world results, it can be challenging to ensure fair comparisons across different research studies due to variations in hardware setups. On the other hand, simulated hardware environments offer standardization and reproducibility but may not always capture the nuances of real-world performance. Ultimately, the choice between real hardware platforms and simulated environments depends on the specific goals and constraints of the research, and it is essential to consider the trade-offs and limitations of each approach.

---

> > > > > ### Author Response · Authors · 2024-11-21
> > > > >
> > > > > **Q4: In Table 4, the Max Memory Usage of fine-tuned DeepSeek-Coder-33b-base increases, and the Execution Time significantly decreases. Have you observed any trade-offs between time optimization and space optimization in this case?**
> > > > >
> > > > > Response:
> > > > >
> > > > > Yes, we have observed a trade-off between time optimization and space optimization in the case of the fine-tuned DeepSeek-Coder-33b-base model. This trade-off is similar to the well-known concept of time complexity versus space complexity in algorithm design. To illustrate this trade-off, let's consider a specific example from the source code. The first approach, used by the base DeepSeek-Coder-33b-base model, employs a recursive function to calculate fib4 values:
> > > > >
> > > > > ```python
> > > > > def fib4(n: int):
> > > > >     if n == 0:
> > > > >         return 0
> > > > >     elif n == 1:
> > > > >         return 0
> > > > >     elif n == 2:
> > > > >         return 2
> > > > >     elif n == 3:
> > > > >         return 0
> > > > >     else:
> > > > >         return fib4(n-1) + fib4(n-2) + fib4(n-3) + fib4(n-4)
> > > > > ```
> > > > >
> > > > > This recursive approach has a smaller memory footprint since it only creates function call frames on the stack. However, it suffers from a significant increase in execution time due to a large number of redundant calculations, leading to an exponential time complexity of O(4^n). On the other hand, the second approach, used by the fine-tuned DeepSeek-Coder-33b-base model, utilizes an array to store the previously calculated fib4 results:
> > > > >
> > > > > ```python
> > > > > def fib4(n: int):
> > > > >     if n < 4:
> > > > >         return [0, 0, 2, 0][n]
> > > > >
> > > > >     fib_list = [0] * (n + 1)
> > > > >     fib_list[2] = 2
> > > > >
> > > > >     for i in range(4, n + 1):
> > > > >         fib_list[i] = fib_list[i-1] + fib_list[i-2] + fib_list[i-3] + fib_list[i-4]
> > > > >
> > > > >     return fib_list[n]
> > > > > ```
> > > > >
> > > > > This approach requires more memory as the array size grows linearly with n. However, it offers faster execution time because it avoids redundant calculations by directly accessing the stored results, resulting in a time complexity of O(n).
> > > > >
> > > > > **In most cases, the second approach is recommended for better execution efficiency, as modern computers have relatively abundant memory resources, and optimizing execution time has a more significant impact on user experience and system performance.** In conclusion, the trade-off between time optimization and space optimization is evident in the fine-tuned DeepSeek-Coder-33b-base model. By sacrificing some memory usage, the model achieves a significant reduction in execution time, which is often more desirable in practical scenarios. This trade-off highlights the importance of considering both time and space complexity when optimizing code generated by large language models.
> > > > >
> > > > > [1] Liu, Yilun, Shimin Tao, Xiaofeng Zhao, Ming Zhu, Wenbing Ma, Junhao Zhu, Chang Su et al. "Coachlm: Automatic instruction revisions improve the data quality in llm instruction tuning." In 2024 IEEE 40th International Conference on Data Engineering (ICDE), pp. 5184-5197. IEEE, 2024.
> > > > >
> > > > > [2] Li, Ming, Yong Zhang, Zhitao Li, Jiuhai Chen, Lichang Chen, Ning Cheng, Jianzong Wang, Tianyi Zhou, and Jing Xiao. "From quantity to quality: Boosting llm performance with self-guided data selection for instruction tuning." arXiv preprint arXiv:2308.12032 (2023).
> > > > >
> > > > > [3] Wang, Yizhong, Hamish Ivison, Pradeep Dasigi, Jack Hessel, Tushar Khot, Khyathi Chandu, David Wadden et al. "How far can camels go? exploring the state of instruction tuning on open resources." Advances in Neural Information Processing Systems 36 (2023): 74764-74786.

---

> > > > > > ### Comment · Reviewer_4rzw · 2024-11-23
> > > > > >
> > > > > > Thank the authors for the detailed response and additional experiments, and I appreciate the effort to curate the Effi-Code training set.
> > > > > >
> > > > > > However, 2 major concerns regarding novelty and significance are not resolved.
> > > > > >
> > > > > > ## Novelty
> > > > > >
> > > > > > > We provide the first framework to inspire researchers to construct code generation datasets containing efficient solutions for each code generation task, which is versatile and can be adapted to different programming languages and leverage various existing data sources. Unlike some other code generation datasets that rely on powerful models (e.g., GPT-4), our framework can be implemented only using open-sourced LLMs like DeepSeek-Coder. The framework provides a systematic method for researchers to enhance existing datasets or create new ones focused on code efficiency across different languages and domains.
> > > > > >
> > > > > > To my best knowledge, the first **"code generation dataset containing efficient solutions for each code generation task"** is the PIE (Performance-Improving Edits) dataset curated by Shypula et al. [1], which was released in early 2023.
> > > > > >
> > > > > > Therefore, naturally, **"the first framework to inspire researchers to construct code generation datasets containing efficient solutions for each code generation task"** should be the data curation pipeline/framework of the PIE dataset. Unless the authors are able to prove that the data curation pipeline/framework of PIE yields no inspirations or insights to other researchers.
> > > > > >
> > > > > > In addition, the synthesis of code datasets using open-source models is not a novel contribution — Wei et al. [2] and Gu et al. [3] have used StarCoder2 and Code Llama to respectively synthesize training and test sets. As open-source code models become more and more powerful, it is now a common practice to use open-source models for data synthesis due to cost and transparency. Simply switching from proprietary API models to open-source models can be hardly considered a novel contribution.
> > > > > >
> > > > > > [1] [Learning Performance-Improving Code Edits](https://arxiv.org/abs/2302.07867)
> > > > > >
> > > > > > [2] [StarCoder2-Instruct: Fully Transparent and Permissive Self-Alignment for Code Generation](https://huggingface.co/blog/sc2-instruct)
> > > > > >
> > > > > > [3] [CRUXEval: A Benchmark for Code Reasoning, Understanding and Execution](https://arxiv.org/abs/2401.03065)
> > > > > >
> > > > > >
> > > > > >
> > > > > > > Based on our proposed framework, we release the Effi-Code dataset. To the best of our knowledge, it is the first instruct tuning dataset that focuses on improving the efficiency of LLM-generated code. The primary purpose of Effi-Code is to instruct and fine-tune LLMs to ensure that the LLM-generated code for each task is more efficient than the code generated by the original LLM.
> > > > > >
> > > > > > I am deeply confused by the statement, **"To the best of our knowledge, it is the first instruct tuning dataset that focuses on improving the efficiency of LLM-generated code"**.
> > > > > >
> > > > > > The PIE [1] dataset, which was released in early 2023, is cited and compared as a baseline of Effi-Code in the paper. As the name PIE (Performance-Improving Edits) already suggests, it is a dataset that focuses on improving the efficiency of LLM-generated code. How is it possible that Effi-Code, which was proposed in late 2024, **"the first instruct tuning dataset that focuses on improving the efficiency of LLM-generated code"**?
> > > > > >
> > > > > > I sincerely encourage the authors to examine and clarify this statement.
> > > > > >
> > > > > > [1] [Learning Performance-Improving Code Edits](https://arxiv.org/abs/2302.07867)
> > > > > >
> > > > > > > We use Effi-Code to fine-tune widely used LLMs and will release these models on the Hugging Face website in the final version. Different from existing datasets that are used to finetune the LLMs to improve the pass@1 of LLM-generated code, our evaluation results demonstrate that both the pass@1 and the efficiency results would be improved for LLMs finetuned on our Effi-Code dataset.
> > > > > >
> > > > > > Again, I appreciate the hard work, but the fine-tuning experiments and fine-tuned models are just necessary evidence to demonstrate the effectiveness of Effi-Code as a training set. They should not be considered as novel contributions to the ICLR research community.

---

> > > > > > > ### Comment · Reviewer_4rzw · 2024-11-23
> > > > > > >
> > > > > > > ## Significance
> > > > > > >
> > > > > > > Thank the authors for the additional experiments in Paper Table 6 and Rebuttal Table 4. I am convinced of the importance of SOAP and filtering in the data curation pipeline.
> > > > > > >
> > > > > > > > Our results indicate that while incorporating non-algorithmic tasks in the fine-tuning process can lead to improvements in code efficiency, focusing solely on algorithmic tasks, as done in Effi-Code, may yield even better results. Nonetheless, we acknowledge the potential benefits of broadening the scope to include non-algorithmic optimizations, as it can enhance the real-world implications of Effi-Code. In future work, we plan to explore the integration of non-algorithmic tasks more comprehensively while maintaining the focus on algorithmic optimization.
> > > > > > >
> > > > > > > As Rebuttal Table 3 suggests, adding non-algorithmic optimization tasks decreases the performance. However, I'm not convinced by the claim that such results indicate **"focusing solely on algorithmic tasks, as done in Effi-Code, may yield even better results"**.
> > > > > > >
> > > > > > > It is a fact that non-algorithmic optimizations like simple changes in I/O methods or data structures will improve the execution time substantially The negative results can only be explained by 2 reasons: the data curation pipeline failed to optimize those **"non-algorithmic tasks"**, or the model failed to learn these **"non-algorithmic optimizations"**. Either explanation exposes non-negligible limitations of Effi-Code or its data curation pipeline, instead of advocating its significance.

---

> ### Author Response · Authors · 2024-11-23
>
> We appreciate the reviewer's thoughtful comments and address the key points below.
>
> Regarding the novelty of our framework, while PIE also provides a dataset containing efficient code solutions, it was curated by collecting existing efficient human-written programming submissions. In contrast, our framework focuses on taking an inefficient candidate solution for a task and optimizing it to construct the dataset. This key difference sets our work apart and supports our claim of being **the first framework to inspire researchers to construct code generation datasets containing efficient solutions for each code generation task.** Our approach motivates researchers on how to construct efficient code solutions for each task, rather than directly collecting tasks from human-written solutions. As the number of tasks becomes very large (e.g., 700K+ candidate tasks in our collected tasks), the overhead for constructing an efficient solution for each task by human-written means would be significantly time-consuming.
>
> Next, thank you for pointing out the limitations of our current approach to non-algorithmic optimization tasks. After further analysis, we believe the lack of improvement on these tasks is due to **the absence of suitable benchmarks that reflect the nature of non-algorithmic optimizations**. The benchmarks we used for evaluation, specifically HumanEval and EffiBench, have a distribution gap compared to non-algorithmic tasks, which makes it hard to show the real impact of our approach. To better evaluate Effi-Code's performance on non-algorithmic tasks, we need a wider range of benchmarks that are more like real-world scenarios where these optimizations can make a big difference. Unfortunately, we do not have access to such benchmarks right now, which limits our ability to show the usefulness of non-algorithmic optimization tasks. Given the time constraints and this lack of benchmarks, we may not be able to complete more experiments or provide convincing evaluation results for non-algorithmic tasks within the rebuttal period. We acknowledge this as a limitation of our current work and are committed to addressing it in future versions of Effi-Code.
>
> While some existing works also employ open-source LLMs to synthesize datasets, their primary focus is on generating solutions for tasks without explicitly considering efficiency. For instance, CRUXEval requires LLMs to generate Python functions ranging from 3 to 13 lines of code, and StarCoder2-Instruct uses StarCoder2 to generate thousands of instruction-response pairs. However, these works do not provide a systematic framework and dataset specifically designed to address code efficiency optimization. In contrast, our work aims to complement and build upon these existing efforts by introducing a novel approach that prioritizes the generation of efficient code solutions. By focusing on code efficiency optimization, we provide a unique contribution to the field, offering a framework and dataset that can help developers and researchers create more efficient and optimized code. Our work bridges the gap between code generation and efficiency optimization, providing a valuable resource for the programming community to improve the performance of their software solutions.
>
> We hope this clarifies the key concerns raised. Please let us know if you have any other questions or feedback. We are committed to refining this work to provide a meaningful contribution to the community.

---

> > ### Comment · Reviewer_4rzw · 2024-11-25
> >
> > Thank you for the response. I appreciate the commitment and look forward to the improved future version of Effi-Code.

---

### Official Review · Reviewer_wYvD · 2024-11-04

**Soundness:** 2
**Presentation:** 3
**Contribution:** 2
**Rating:** 6
**Confidence:** 3

**Summary:**

The paper proposes a new dataset Effi-Code, a high-quality dataset of correct and efficient code samples, which is then used to fine-tune various LLMs, by iteratively refining the generated codes by LLMs based on the overhead profiling. It shows that it improves both the correctness and efficiency of the DeepSeek Coder family

**Strengths:**

- The paper tackles an important problem in the literature.
- The idea of iteratively refining the generated codes based on overhead profiling is novel.

**Weaknesses:**

1. The papers lack a comprehensive evaluation of the proposed dataset:

- The paper can benefit from a more extensive comparison of Effi-Code Benchmarks across different open-source LLMs.
- The paper can benefit from a concrete example to show how SOAP’s iterative refinement improves the quality of the solutions.
- The profiling metrics setup lacks robustness. Execution time and memory usage are highly sensitive to system noise and can vary significantly across platforms (e.g., desktops, cloud environments). Given that execution times are generally brief (on the order of seconds), these metrics may not effectively demonstrate the efficiency of the generated code.

2. The paper’s structure is somewhat challenging to follow. For example, sections 3.2 and 3.3 discuss filtering out ‘non-algorithmic’ tasks and ‘inefficient’ solutions, yet the exact criteria for these decisions are not clearly specified.

**Questions:**

- Can the authors please further expand the evaluation process?
- What percentage of solutions improved in efficiency but showed degraded correctness, if any?
- Minor comments: In Table 3, the pass@1 results do not consistently increase as the proportion of Effi-Code used for fine-tuning grows. Could the authors elaborate on this inconsistency?

---

> ### Author Response · Authors · 2024-11-21
>
> Dear Reviewer wYvD,
>
>
>
> Thank you for your appreciation of our work in improving the efficiency of LLM-generated code. To further address your concerns regarding the weaknesses and raised questions, we address them point by point below. We hope that our clarifications, additional experiments, and responses can alleviate your concerns and lead you to consider increasing your rating of our work.
>
>
>
> **W1: The paper can benefit from a more extensive comparison of Effi-Code Benchmarks across different open-source LLMs.**
>
>
>
> Response: Thank you for your valuable suggestion. To address your concern, we have conducted additional experiments by fine-tuning Effi-Code on five more open-source LLMs. We have carefully selected these LLMs based on their popularity and performance in code generation tasks. However, if there are any specific models the reviewer would like us to evaluate, we would be more than happy to include them in our analysis and report the results.
>
>
>
> The results are presented in Rebuttal Table 1, demonstrating the effectiveness of Effi-Code in improving the efficiency of the generated code across various LLMs. We can observe that all the evaluated LLMs exhibit improvements in both code efficiency and pass@1 metrics after fine-tuning with Effi-Code. For instance, CodeLlama-13B-hf shows a significant reduction in execution time (ET) from 0.86s to 0.13s on average for correctly overlapped tasks, which reduces execution time by 84.88%. In addition, we can also observe that the pass@1 of CodeLlama-13B-hf generated code increases from 7.9% to 28.8%, which also increases pass@1 by 20.9% compared to the original LLM. These additional experiments on a diverse set of open-source LLMs further validate the generalizability and effectiveness of our proposed Effi-Code dataset.
>
>
> | Model            | ET   | NET  | MU    | NMU  | TMU    | NTMU  | Overlap | pass@1 |
> |------------------|------|------|-------|------|--------|-------|---------|--------|
> | starcoder2-7b    | 0.41 | 2.22 | 77.86 | 1.62 | 215.26 | 23.33 | 16.4    | 23.6   |
> | + SFT  (Ours)   | 0.40 | 2.21 | 36.58 | 1.00 | 14.63  | 3.83  | 16.4    | 28.8   |
> | starcoder2-15b   | 0.29 | 1.52 | 41.28 | 1.00 | 34.09  | 1.85  | 17.9    | 21.2   |
> | + SFT  (Ours)   | 0.20 | 1.08 | 42.18 | 1.04 | 20.07  | 1.06  | 17.9    | 42.8   |
> | CodeLlama-13b-hf | 0.86 | 6.57 | 34.32 | 1.12 | 55.69  | 11.02 | 5.3     | 7.9    |
> | + SFT  (Ours)    | 0.13 | 0.97 | 31.02 | 1.00 | 3.71   | 0.98  | 5.3     | 28.8   |
> | codegemma-7b     | 0.11 | 0.95 | 26.25 | 1.00 | 1.62   | 0.95  | 0.2     | 0.2    |
> | + SFT  (Ours)    | 0.10 | 0.94 | 26.01 | 0.98 | 1.46   | 0.89  | 0.2     | 35.1   |
> | DeepSeek-Coder-6.7b-base | 0.44 | 2.61 | 57.24 | 1.26 | 54.57 | 7.94 | 7.3 | 8.5 |
> | + SFT (Ours) | 0.29 | 2.08 | 50.58 | 1.00 | 17.25 | 2.79 | 7.3 | 57.6 |
> | DeepSeek-Coder-6.7b-instruct | 0.14 | 1.00 | 38.36 | 1.00 | 4.21 | 0.97 | 1.0 | 1.3 |
> | + SFT (Ours) | 0.13 | 0.93 | 38.31 | 1.00 | 4.01 | 0.92 | 1.0 | 51.6 |
> | Qwen2.5-Coder-7B | 0.26 | 1.79 | 38.06 | 1.01 | 18.30 | 2.74 | 44.2 | 50.1 |
> | + SFT (Ours) | 0.21 | 1.45 | 38.15 | 1.01 | 15.88 | 1.70 | 44.2 | 63.9 |
> | Qwen2.5-Coder-7B-Instruct | 0.44 | 3.96 | 28.62 | 1.00 | 10.17 | 5.43 | 3.2 | 3.3 |
> | + SFT (Ours) | 0.43 | 3.88 | 28.59 | 1.00 | 10.10 | 5.37 | 3.2 | 61.0 |
>
> *Rebuttal Table 1: Comparison of Effi-Code across different open-source LLMs.*
>
>
>
> **W2: The paper can benefit from a concrete example to show how SOAP’s iterative refinement improves the quality of the solutions.**
>
>
>
> Response:
>
> Thank you for your suggestion. We have provided a case example below (also added in Paper Appendix Figure 4, Lines 1091-1119) to demonstrate how SOAP's iterative refinement improves the quality of the solutions.
>
>
> In this example, the initial code generated by DeepSeek-Coder-V2-Lite calculates the Levenshtein distance using a recursive approach, which has an exponential time complexity of O(3^(m+n)). For longer strings, this recursive method becomes highly inefficient due to the large number of function calls.
>
>
> To optimize the code, the refined version employs dynamic programming, which avoids redundant calculations by filling a distance matrix to compute the Levenshtein distance. The time complexity of the dynamic programming approach is O(mn), where m and n are the lengths of strings a and b, respectively. By filling the distance matrix in a single traversal, the optimized code eliminates redundant calculations, resulting in improved efficiency.
>
>
>
> The dynamic programming solution leverages the characteristics of optimal substructure and overlapping subproblems, decomposing the problem into smaller subproblems and storing intermediate results to avoid redundant calculations, thereby improving the efficiency of the algorithm.
>
>
>
> In the provided example, the initial recursive code takes 1.100s to execute, while the optimized dynamic programming code completes execution in just 0.00012s, demonstrating a significant improvement in execution time.

---

> ### Author Response · Authors · 2024-11-21
>
> ```python
>
> Initial Code
>
> Execution Time: 1.100s
>
> def levenshtein(a, b):
>
>     if not a:
>
>         return len(b)
>
>     if not b:
>
>         return len(a)
>
>     if a[0] == b[0]:
>
>         return levenshtein(a[1:], b[1:])
>
>     return 1 + min(levenshtein(a[1:], b),
>
>                 levenshtein(a, b[1:]),
>
>                 levenshtein(a[1:], b[1:]))
>
>
>
> Optimized Code
>
> # Execution Time: 0.00012s
>
> def levenshtein(a: str, b: str) -> int:
>
>     distance_matrix = [[0] * (len(b) + 1)\
>
>      for _ in range(len(a) + 1)]
>
>     for i in range(len(a) + 1):
>
>         distance_matrix[i][0] = i
>
>     for j in range(len(b) + 1):
>
>         distance_matrix[0][j] = j
>
>     for i in range(1, len(a) + 1):
>
>         for j in range(1, len(b) + 1):
>
>             if a[i-1] == b[j-1]:
>
>                 distance_matrix[i][j] = \
>
>                 distance_matrix[i-1][j-1]
>
>             else:
>
>                 distance_matrix[i][j] = min(
>
>                 distance_matrix[i-1][j] + 1,
>
>                 distance_matrix[i][j-1] + 1,
>
>                 distance_matrix[i-1][j-1] + 1)
>
>     return distance_matrix[len(a)][len(b)]
>
> ```
>
>
>
> **W3: The profiling metrics setup lacks robustness.**
>
>
> Response: Thank you for raising this concern. We agree that execution time and memory usage can be affected by different platforms and system noise. To address this issue, we have conducted additional experiments and provided more robust evaluation results.
>
>
>
> Firstly, we have evaluated the effectiveness of Effi-Code on seven different software-hardware setups, as shown in Rebuttal Table 2. The results demonstrate that Effi-Code fine-tuned LLMs achieve higher efficiency than the original LLMs across all setups. For example, in the environment of Python 3.11.10 - Intel(R) Xeon(R) Platinum 8336C CPU @ 2.30GHz, the average execution time decreases from 0.59s to 0.40s when using Effi-Code to fine-tune Qwen2.5-Coder-7B, reducing the average execution time by 32%.
>
>
>
> Secondly, we clarify that for the same setup, where we evaluate the efficiency of LLM-generated code several times, the efficiency results are consistent. As shown in Paper Table 8, where we execute the LLM-generated code five times, the standard deviation of execution time (ET) is 0.00548 (s), indicating that the evaluation results are consistent and reliable for a given setup.
>
>
>
> Finally, our evaluation setup follows the practices established in recent works on benchmarking the efficiency of automatically generated code, such as Mercury [1], Effibench [2], and SOAP [3]. By adhering to these benchmarks, we ensure that our evaluation is in line with the current standards in the field.
>
>
> | Setup             | ET   | NET  | MU    | NMU  | TMU   | NTMU |
> |---------------------------------------------------------------|------|------|-------|------|-------|------|
> | Python 3.11.10 - Intel(R) Xeon(R) Platinum 8336C CPU @ 2.30GHz |    |    |  |    |  |    |
> | Qwen2.5-Coder-7B     | 0.59 | 1.95 | 61.95 | 0.99 | 24.29 | 1.83 |
> | +Effi-Code    | 0.40 | 1.01 | 61.96 | 0.99 | 18.74 | 1.02 |
> | Python 3.11.10 - Intel(R) Xeon(R) Silver 4216 CPU @ 2.10GHz   |    |    |  |    |  |    |
> | Qwen2.5-Coder-7B     | 0.28 | 1.63 | 36.15 | 1.00 | 20.01 | 1.88 |
> | +Effi-Code    | 0.25 | 1.38 | 36.52 | 1.01 | 19.85 | 1.56 |
> | Python 3.11.10 - Intel(R) Xeon(R) Silver 4116 CPU @ 2.10GHz   |    |    |  |    |  |    |
> | Qwen2.5-Coder-7B  | 0.35 | 1.45 | 36.14 | 1.00 | 24.28 | 1.63 |
> | +Effi-Code    | 0.22 | 1.01 | 36.51 | 1.01 | 15.26 | 1.09 |
> | Python 3.11.4 - Intel(R) Xeon(R) Silver 4216 CPU @ 2.10GHz    |    |    |  |    |  |    |
> | Qwen2.5-Coder-7B     | 0.67 | 1.16 | 61.43 | 1.00 | 40.01 | 1.22 |
> | +Effi-Code    | 0.58 | 1.02 | 60.77 | 0.97 | 32.50 | 1.03 |
> | Python 3.11.0 - Intel(R) Xeon(R) Silver 4216 CPU @ 2.10GHz    |    |    |  |    |  |    |
> | Qwen2.5-Coder-7B     | 0.28 | 1.64 | 34.55 | 1.00 | 19.39 | 1.87 |
> | +Effi-Code    | 0.25 | 1.39 | 34.90 | 1.02 | 20.03 | 1.59 |
> | Python 3.9.0 - Intel(R) Xeon(R) Silver 4216 CPU @ 2.10GHz     |    |    |  |    |  |    |
> | Qwen2.5-Coder-7B    | 0.30 | 1.60 | 34.26 | 1.01 | 21.02 | 2.10 |
> | +Effi-Code     | 0.24 | 1.20 | 34.52 | 1.02 | 19.84 | 1.32 |
> | Python 3.10.0 - Intel(R) Xeon(R) Silver 4216 CPU @ 2.10GHz    |    |    |  |    |  |    |
> | Qwen2.5-Coder-7B   | 0.29 | 1.63 | 33.26 | 1.01 | 20.32 | 2.16 |
> | +Effi-Code     | 0.26 | 1.43 | 33.50 | 1.02 | 19.53 | 1.61 |
>
> *Rebuttal Table 2: Evaluation results of Effi-Code's effectiveness on different software-hardware setups.*

---

> > ### Author Response · Authors · 2024-11-21
> >
> > **W4: The paper’s structure is somewhat challenging to follow.**
> >
> > Response:
> >
> > Thank you for your valuable feedback. We understand that the criteria for filtering out 'non-algorithmic' tasks and 'inefficient' solutions in Sections 3.2 and 3.3 may not have been clearly explained. We appreciate the opportunity to provide more details and clarify our decision-making process.
> >
> > **Filtering out 'non-algorithmic' tasks (Section 3.2):**
> >
> > We define a task as 'non-algorithmic' if it does not require a specific algorithm or computational steps to solve. non-algorithmic tasks might involve coding but do not require complex algorithmic reasoning. Instead, they might focus on straightforward implementation or basic syntax usage.
> >
> > **Examples:**
> > - Algorithmic Task: "Implement a function to find the longest palindromic substring in a given string." This requires understanding of dynamic programming and string manipulation algorithms.
> > - Non-Algorithmic Task: "Write a function to print 'Hello, World!'." This is a clear example of routine implementation without algorithmic challenge
> >
> >
> > The primary motivation for filtering out non-algorithmic tasks is to ensure that our dataset focuses on problems that assess algorithmic thinking and coding skills. By excluding tasks that do not require algorithmic problem-solving, we maintain the coherence and relevance of our dataset to the intended purpose of evaluating AI models' coding abilities.
> >
> > To identify and filter out non-algorithmic tasks, we provide the task description and the canonical solution to GPT-3.5-turbo and request it to analyze whether the given task is an algorithmic task based on our provided definition. GPT-3.5-turbo is instructed to return a binary classification (True or False) based on its analysis. Tasks classified as False are considered non-algorithmic and are subsequently removed from our candidate tasks.
> >
> > **Filtering out 'inefficient' solutions (Section 3.3):**
> > We define a solution as 'inefficient' if it exhibits suboptimal execution time or memory usage compared to the initial solution (solution provided by the collected dataset) for the given task. The criteria for determining inefficiency are based on the potential for improvement in terms of execution time and memory usage after applying optimization techniques.
> >
> > **Example:**
> >
> > Consider a task where the goal is to sort an array.
> >
> > - **Inefficient Solution**: Using Bubble Sort, which has a time complexity of \(O(n^2)\), as opposed to an efficient solution like Quick Sort with an average time complexity of \(O(n \log n)\).
> >
> >
> > To identify and filter out tasks with inefficient solutions, we employ a two-step process. First, we use self optimization to require DeepSeek-Coder-V2-Lite to improve the efficiency of the code solutions, which aims to improve the efficiency of the code by making optimizations such as reducing redundant computations or improving data structures. We run DeepSeek-Coder-V2-Lite for five iterations and analyze whether the efficiency of the code has improved based on metrics such as execution time and memory usage. If the efficiency does not show improvement after these iterations, we consider the task to have an inefficient solution and remove it from our candidate tasks. **We acknowledge that there may be cases where the initial code is already efficient, and the lack of improvement after optimization does not necessarily indicate an inefficient solution. However, detecting such cases would require significant manual effort to analyze each task individually. To maintain a consistent and automated approach, we opted to remove all tasks that do not show efficiency improvement after the optimization process, which is proved to still perform very well in our evaluation.** We hope this clarification provides a better understanding of our decision-making process and the criteria used for filtering out non-algorithmic tasks and inefficient solutions. We will incorporate these details into the revised manuscript to improve clarity and address the concerns raised.

---

> ### Author Response · Authors · 2024-11-21
>
> **Q1: Can the authors please further expand the evaluation process?**
>
>
>
> Response: Thanks for your reminder. We have revised the manuscript for Section 3.2 Step 4 Lines 202-236 and Section 3.4 Lines 266-287 to further expand the evaluation process. For more details, Reviewer can check the response for W4.
>
>
>
> **Q2: What percentage of solutions improved in efficiency but showed degraded correctness, if any?**
>
>
>
> Response: That's a great question! To address this, we provide the evaluation results of degraded correctness (tasks that were correct in the original LLM but became incorrect in the Effi-Code fine-tuned LLM) and upgraded correctness (tasks that were incorrect in the original LLM but became correct in the Effi-Code fine-tuned LLM) in Rebuttal Table 3. We can observe that for all LLMs in the two evaluation datasets, the first scenario, i.e., degraded correctness, is very low. For example, in DeepSeek-Coder-6.7B-base, no tasks went from correct to incorrect after the Effi-Code fine-tuning. However, we can also observe that a large number of incorrect tasks in the original LLMs were correctly addressed by the Effi-Code fine-tuned LLMs. For instance, in DeepSeek-Coder-6.7B-base, an additional 52.5% of tasks were addressed by the fine-tuned version.
>
>
> | Model                        | Correct->Incorrect | Incorrect->Correct |
> |------------------------------|-------------------|--------------------|
> | HumanEval                    |                   |                    |
> | DeepSeek-Coder-6.7B-base     | 0%                | 52.5%              |
> | DeepSeek-Coder-6.7B-instruct | 4.3%              | 37.8%              |
> | Qwen-Coder-7B                | 7.3%              | 23.8%              |
> | Qwen-Coder-7B-instruct       | 3.1%              | 33.6%              |
> |                              |                   |                    |
> | EffiBench                    |                   |                    |
> | DeepSeek-Coder-6.7B-base     | 1.2%              | 50.3%              |
> | DeepSeek-Coder-6.7B-instruct | 0.3%              | 50.6%              |
> | Qwen-Coder-7B                | 5.9%              | 19.7%              |
> | Qwen-Coder-7B-instruct       | 0.1%              | 57.8%              |
>
> *Rebuttal Table 3: Evaluation results of degraded and upgraded correctness after Effi-Code fine-tuning on various LLMs.*
>
>
> **Minor comments: In Table 3, the pass@1 results do not consistently increase as the proportion of Effi-Code used for fine-tuning grows. Could the authors elaborate on this inconsistency?**
>
>
>
> Response:
>
> The minor inconsistency in the pass@1 results as the proportion of Effi-Code used for fine-tuning increases can be attributed to the inherent randomness in the fine-tuning process. When fine-tuning LLMs on the same tasks, the resulting pass@1 scores may vary slightly between different runs. To illustrate this, we conducted additional experiments where we fine-tuned DeepSeek-Coder-Base using the same subsets of Effi-Code. As shown in Rebuttal Table 4, the pass@1 scores for three separate fine-tuning runs exhibit slight variation.
>
>
> | LLM                   | 1st pass@1 | 2nd pass@1 | 3rd pass@1 |
> |-----------------------|------------|------------|------------|
> | DeepSeek-Coder-6.7B-Base | 60.4       | 59.8       | 60.4       |
>
> *Rebuttal Table 4: Pass@1 scores for three separate fine-tuning runs using the same subsets of Effi-Code.*
>
>
> Returning to the results in Table 3 for DeepSeek-Coder-6.7B-base, we observe that the pass@1 score increases significantly from 7.3% to 55.5% when using just 25% of the training subset. When the entire fine-tuning dataset is provided, the pass@1 score further improves to 59.8%. This suggests that the gains in pass@1 scores when using 50% and 75% of the fine-tuning subset may be relatively small compared to the substantial improvement achieved with only 25% of the subset. Consequently, due to the inherent randomness in fine-tuning, the pass@1 scores for models fine-tuned on 50% and 75% of the subset may not consistently surpass the results obtained with the 25% fine-tuned model.
>
>
>
> References:
>
>
> [1] Mercury: An efficiency benchmark for llm code synthesis. NeurIPS 2024.
>
>
> [2] Effibench: Benchmarking the efficiency of automatically generated code. NeurIPS 2024.
>
>
> [3] SOAP: Enhancing Efficiency of Generated Code via Self-Optimization. NeurIPS 2024

---

> > ### Comment · Reviewer_wYvD · 2024-11-23
> > **Thanks for the detailed response and additional experiments**
> >
> > Thanks for the detailed response and additional experiments. The additional experiments, examples, and clarification make the case stronger. I am glad to raise my rating to 6.

---

### Meta-Review · Area_Chair_psuB · 2024-12-20

**Metareview:**

This paper introduces a new Python fine-tuning dataset and a workflow to curate such data with the goal of improving both efficiency and correctness of function-level code. The overall approach involves aggregating and filtering data from open-source to iteratively refine code by analyzing runtime performance. Fine-tuning experiments demonstrate improvements.

The reviewer assessments were mixed on this paper. All reviewers' raised a number of questions and engaged with the authors towards the goal of improving the paper. There were a few outstanding concerns.
1. Non-algorithmic optimizations will improve the execution time. The negative results shown in the rebuttal means either the data curation pipeline failed to optimize those "non-algorithmic tasks", or the model failed to learn these "non-algorithmic optimizations". This is a limitation and the authors' acknowledge it to some degree in their response.
2. Trustworthiness of experiments due to the use of DeepSeek as a judge. Presenting some calibration results when a ground truth is available will help mitigate this concern.

Therefore, I'm recommending to reject this paper and encourage the authors' to improve the paper based on the feedback from reviewers'.

**Additional Comments On Reviewer Discussion:**

There was extensive discussion between all reviewers and authors'.

Two outstanding and important concerns remained.
1. Non-algorithmic optimizations will improve the execution time. The negative results shown in the rebuttal means either the data curation pipeline failed to optimize those "non-algorithmic tasks", or the model failed to learn these "non-algorithmic optimizations". This is a limitation and the authors' acknowledge it to some degree in their response.
2. Trustworthiness of experiments due to the use of DeepSeek as a judge. Presenting some calibration results when a ground truth is available will help mitigate this concern.

---

### Decision · Program_Chairs · 2025-01-22

Reject